# Scaling Laws for Reward Model Overoptimization in Direct Alignment Algorithms

**Rafael Rafailov***
Stanford University
rafailov@cs.stanford.edu

**Yaswanth Chittepu***
UMass Amherst
ychittepu@umass.edu

**Ryan Park***
Stanford University
rypark@stanford.edu

**Harshit Sikchi***
UT Austin
hsikchi@utexas.edu

**Joey Hejna***
Stanford University
jhejna@cs.stanford.edu

**W. Bradley Knox**
UT Austin
bradknox@cs.utexas.edu

**Chelsea Finn**
Stanford University
cbfinn@cs.stanford.edu

**Scott Niekum**
UMass Amherst
sniekum@cs.umass.edu

## Abstract

Reinforcement Learning from Human Feedback (RLHF) has been crucial to the recent success of Large Language Models (LLMs), however, it is often a complex and brittle process. In the classical RLHF framework, a reward model is first trained to represent human preferences, which is in turn used by an online reinforcement learning (RL) algorithm to optimize the LLM. A prominent issue with such methods is *reward over-optimization* or *reward hacking*, where performance as measured by the learned proxy reward model increases, but true quality plateaus or even deteriorates. Direct Alignment Algorithms (DAAs) like Direct Preference Optimization have emerged as alternatives to the classical RLHF pipeline by circumventing the reward modeling phase. However, although DAAs do not use a separate proxy reward model, they still commonly deteriorate from over-optimization. While the so-called reward hacking phenomenon is not well-defined for DAAs, we still uncover similar trends: at higher KL budgets, DAA algorithms exhibit similar degradation patterns to their classic RLHF counterparts. In particular, we find that DAA methods deteriorate not only across a wide range of KL budgets but also often before even a single epoch of the dataset is completed. Through extensive empirical experimentation, this work formulates and formalizes the reward over-optimization or hacking problem for DAAs and explores its consequences across objectives, training regimes, and model scales.

## 1 Introduction

Recent advancements in Large Language Models (LLMs) have broadened their capabilities significantly, enabling applications in code generation, mathematical reasoning, tool use, and interactive communication. These improvements have popularized LLMs across various domains. Reinforcement Learning from Human Feedback (RLHF) has been instrumental in these advances and is now integral to sophisticated LLM training regimes [10, 55]. Before alignment, LLMs, trained on vast text corpses to predict subsequent tokens [45, 8] are often unwieldy and hard to use. Today, leading LLMs incorporate variants of the RLHF framework [14, 69, 36] to align them with human intent, which

*Equal Contribution, Dice Rolling

38th Conference on Neural Information Processing Systems (NeurIPS 2024).

generally involves a multi-stage process. Specifically, users evaluate model responses to assorted prompts in order to train a reward model that encapsulates human preferences [10, 55, 72, 5, 62]. Then, the refined LLM maximizes the expected learned reward function using a reinforcement learning (RL) algorithm [50, 1, 65]. Despite its efficacy, this procedure is complex and computationally intensive, particularly in its latter stages.

Goodhart's Law [25, 11], that "when a measure becomes a target, it ceases to be a good measure", has often been cited as a core shortcoming of RLHF. Standard RLHF methods optimize a learned, but imperfect reward function which ends up amplifying the reward model's shortcomings. Empirically, this phenomenon was first extensively characterized by Gao et al. [21], who coined the term "reward over-optimization", and has been seen consistently in recent findings [62, 16, 14]. While reward over-optimization has been studied in the context of the aforementioned RLHF procedure, recent contemporary methods for aligning LLMs circumvent the reward learning procedure, necessitating a new characterization of the over-optimization phenomena.

This new broad class of algorithms, which we refer to as Direct Alignment Algorithms (DAAs), bypass the traditional RLHF pipeline by re-parameterizing the reward model directly through the optimal policy derived during the reinforcement learning phase. DAA methods, like Direct Preference Optimization [46], have gained popularity [14, 28] as they often reduce computational demands. Yet, despite not fitting a reward function, DAAs still exhibit over-optimization trends similar to those of traditional RLHF methods using a learned reward function. In some sense, this is puzzling: DAAs can be viewed as simply learning a reward function with supervised learning from which the optimal policy is deterministically mapped, however more seems to be at play than simple supervised learning.

In this work, we investigate the over-fitting phenomena present in DAA algorithms through extensive experimentation. First, we unify a number of different recent methods [46, 68, 4] under the DAA framework. Then, across different model scales and hyper-parameters, we show that DAAs exhibit a type of reward over-optimization consistent with that previously observed in RLHF [21]. Specifically, we find that at different KL-divergence budgets DAAs exhibit degradation patterns similar to those found in RLHF. Interestingly, we also find that performance within a single epoch is not always as consistent as expected for DAAs. Finally, we explain why this happens by appealing to the under-constrained nature of the optimization problem used in DAAs.

## 2 Preliminaries

In this section, we first outline the core components of the standard RLHF pipeline [72, 55, 5, 41]). Then, we examine prior literature to characterize the reward over-optimization exhibited by standard RLHF methods. Finally, we provide a unifying view of direct alignment algorithms (DAAs) which will guide our analysis of their training dynamics in the next section.

### 2.1 Reinforcement Learning From Human Feedback

The standard RLHF pipeline consists of three distinct stages with the goal of aligning the LLM with human preferences.

**Supervised Fine Tuning (SFT)**: First, a dataset of prompts $x$ and high-quality answers $y$ are used to train an LLM for instruction following via maximum likelihood estimation over next-tokens. We refer to the resultant model as $\pi_{\text{SFT}}(y|x)$ and consider the entire prompt and answer strings to be single variables.

**Reward Modeling**: Second, the SFT model $\pi_{\text{SFT}}(y|x)$ is used to learn a reward function over human preferences. Specifically, the SFT model is queried to produce pairs of answers $(y_1, y_2) \sim \pi_{\text{SFT}}(y|x)$, for every prompt $x$ in a dataset. Then, users select their preferred answers, resulting in ranking $y_w \succ y_l \mid x$ where $y_w$ and $y_l$ are the preferred and dispreferred answers respectively. Typically, user rankings are assumed to be distributed according to the Bradley-Terry (BT) model [7]

$$p(y_1 \succ y_2 \mid x) = \frac{\exp\left(r(x, y_1)\right)}{\exp\left(r(x, y_1)\right) + \exp\left(r(x, y_2)\right)} = \sigma(r(x, y_1) - r(x, y_2)) \tag{1}$$

where the preference distribution $p$ results from an unobserved latent reward $r(x, y)$, and $\sigma$ is the logistic function. Given this model and a dataset of rankings, denoted $\mathcal{D} = \left\{x^{(i)}, y_w^{(i)}, y_l^{(i)}\right\}_{i=1}^{N}$, we

can train a parameterized model $r_\phi(x, y)$ to predict the unobserved reward using maximum likelihood estimation. This yields the following loss function,

$$\mathcal{L}_{\text{rew}}(r_\phi) = -\mathbb{E}_{(x,y_w,y_l)\sim\mathcal{D}}\big[\log\sigma(r_\phi(x, y_w) - r_\phi(x, y_l))\big]. \tag{2}$$

**Reinforcement Learning (RL)**: The final stage of the standard RLHF pipeline uses the learned reward model $r_\phi(x, y)$ to update the LLM $\pi_\theta$ with an on-policy RL algorithm like PPO [50], optimizing the model to provide responses more preferred by human raters. The most common objective is

$$\max_{\pi_\theta} \mathbb{E}_{x\sim\mathcal{D}, y\sim\pi_\theta(.|x)}\big[r_\phi(x, y)\big] - \beta\mathbb{D}_{\text{KL}}\big[\pi_\theta(y \mid x) \mid\mid \pi_{\text{ref}}(y|x)\big] \tag{3}$$

which enforces a Kullback-Leibler (KL) divergence penalty with a reference distribution $\pi_{\text{ref}}(y|x)$ (usually taken to be $\pi_{\text{SFT}}(y|x)$) to prevent the LLM $\pi_\theta$ from straying too far from its initialization. Thus, the hyper-parameter $\beta$ directly trades off exploiting the reward function and deviating from $\pi_{\text{ref}}(y|x)$.

### 2.2 Reward Exploitation in RLHF

Unfortunately, repeating the above procedure without careful tuning of the RL phase can lead to disastrous performance. This is because in the context of RLHF the LLM policy is optimizing the surrogate reward estimate $r_\phi(x, y)$ and not the true reward function as is often the case in other domains. Thus, prior works have observed that while the LLM's expected reward according to eq. (3) increases the actual quality of the model's outputs can decrease [54, 43, 9, 34]. This particular instantiation of the reward exploitation or hacking problem [3] is often referred to as reward "over-optimization" in RLHF literature and has been studied empirically in both controlled experiments [21] and user studies [14]. There are two prevailing explanations for why this phenomenon occurs.

**1. OOD Robustness:** In the classical RLHF pipeline, the RL objective (eq. (3)) is optimized using on-policy samples from $\pi_\theta$. This means that the reward function is continuously queried using unseen model samples which are potentially out-of-distribution. Beyond the support of the reward modeling distribution, $r_\phi$ may assign high rewards to sub-par responses, leading the policy to believe it is doing well when it may not be. While the KL-regularization term is designed to prevent the model from drifting too far out of distribution, this term alone has proven inadequate to prevent reward hacking [21].

**2. Reward Mis-specification.** Learned reward functions may exhibit spurious correlations that cause them to prefer unintended behaviors. While this issue is not at the forefront of LLM research, it is known to be pervasive in RL [43, 34]. Most efforts to address these problems exist at the intersection of robustness and offline RL literature [13, 67, 16] and use measures of epistemic uncertainty to penalize the predicted reward.

### 2.3 Direct Alignment Algorithms

Due to its complex multi-step nature, recent works have sought alternatives to the classic RLHF pipeline. A new class of algorithms, which we broadly classify as Direct Alignment Algorithms (DAAs), directly update the LLM's policy $\pi_\theta$ using user feedback instead of fitting a reward function to it and then employing an RL algorithm. Perhaps the most known example is Direct Preference Optimization (DPO). DPO, as well as other DAAs, are derived using the closed form solution to the RLHF objective in eq. (3) [71], $\pi^*(y|x) \propto \pi_{\text{ref}}(y|x)e^{r(x,y)/\beta}$, where $r(x, y)$ is the ground-truth reward. By isolating $r(x, y)$ in this relationship and substituting it into the reward optimization objective in eq. (2), we arrive at a general objective that allows us to train the LLM directly using feedback data:

$$\mathcal{L}_{\text{DAA}}\left(\pi_\theta; \pi_{\text{ref}}\right) = \mathbb{E}_{(x,y_w,y_l)\sim\mathcal{D}}\left[g\left(\beta\log\frac{\pi_\theta\left(y_w \mid x\right)}{\pi_{\text{ref}}\left(y_w \mid x\right)} - \beta\log\frac{\pi_\theta\left(y_l \mid x\right)}{\pi_{\text{ref}}\left(y_l \mid x\right)}\right)\right] \tag{4}$$

where $g$ is a convex loss function. Using $g(x) = -\log\sigma(x)$ coincides with the standard Bradley-Terry model and the original DPO objective. Other methods choose different loss functions: IPO [4] uses the quadratic objective $g(x) = (x - 1)^2$ and SLiC-HF [68, 38] uses the hinge loss $g(x) = \max(0, 1 - x)$. Additional objectives were also considered in [59], but due to limited computational resources, we focus on the three objectives outlined above.

Crucially, the DAA approach allows us to recover the optimal policy using a straightforward classification loss without the need for learning a reward function, on-policy sampling, or RL, which can be

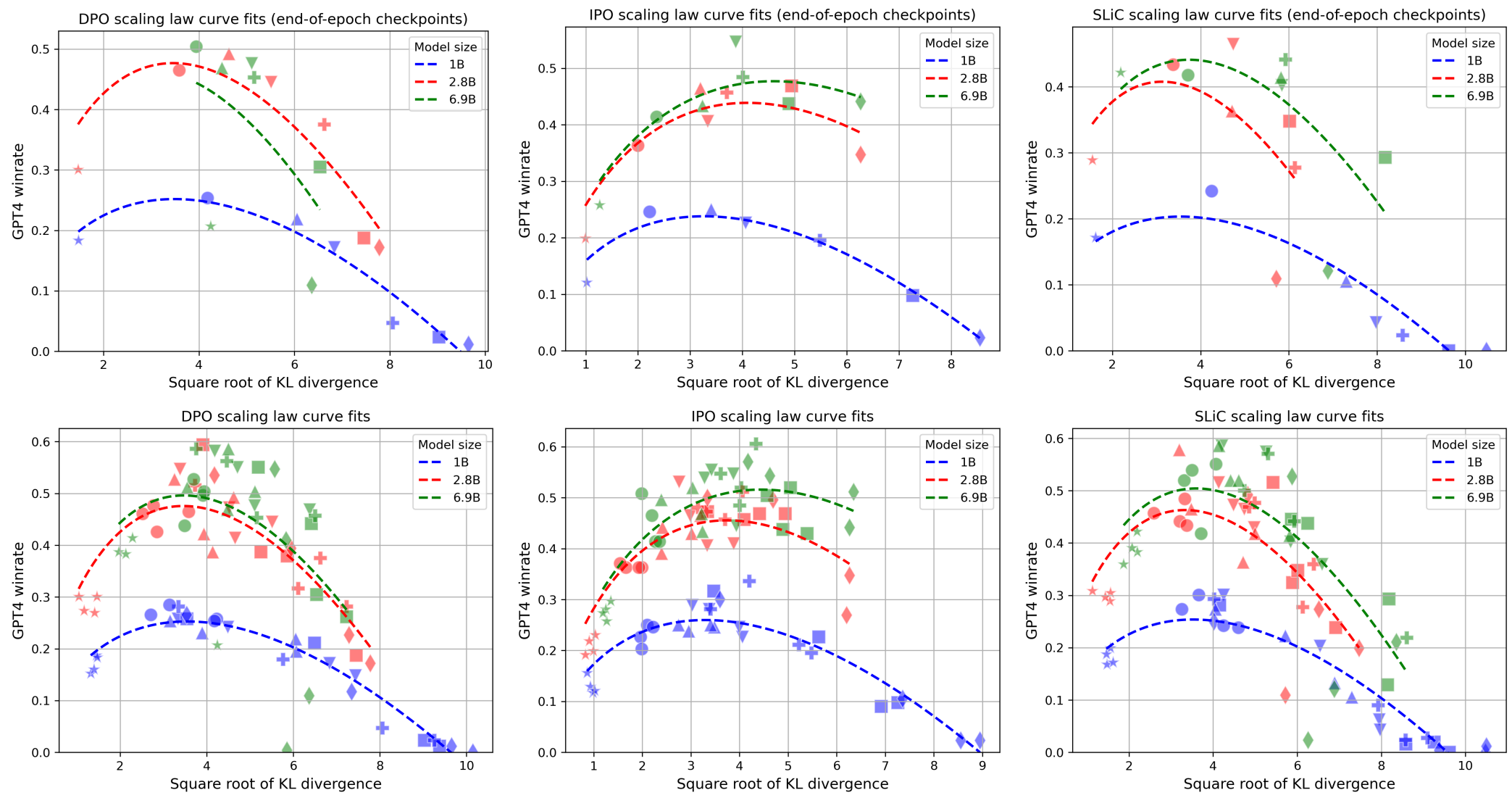

Figure 1: Results on over-optimization in Direct Alignment Algorithms for DPO, IPO and SLiC. Results show model win-rates over the dataset summary on an evaluation set of prompts as judged by GPT-4. The top row shows the final performance after 1 epoch of training, while the second row also includes 4 intermediate checkpoints as well. The fitted dotted curves utilize scaling laws from [21] applied to direct alignment, with GPT4 winrates taking the place of the gold reward model score.

notoriously difficult to tune and computationally expensive. Because of this, DAAs have emerged as a popular alternative. However, just like classical RLHF methods, DAAs exhibit strong over-fitting and even reward-hacking like behaviors. For example, Park et al. [44] show that LLMs trained with DPO generate responses with increasing length throughout the course of training, but do not improve in ground-truth win-rate after a certain point. Since DAAs do not explicitly learn a reward function, it is unclear how "reward-overoptimization" fits into the picture. In this work, we aim to shed some light on this phenomenon in DAAs.

## 3 Empirical Analysis of Overoptimization in DAAs

First, we examine the over-optimization problem in DAAs and compare it to those observed in traditional RLHF methods. All our experiments are carried out using the Reddit TL;DR summarization dataset [55] and the Pythia family of Large Language Models [6]. Additional plots illustrating similar over-optimization trends for Direct Alignment Algorithms on the Gemma2-2b model [61] and the Anthropic Helpfulness-Harmlessness dataset [5] are provided in Appendix F

### 3.1 Evaluating Model-Overoptimization

In our first set of experiments, we evaluate the reward model over-optimization phenomenon. We evaluate three training objectives DPO, IPO, and SLiC using seven $\beta$ parameters, representing different KL budgets at three model sizes - 1B, 2.8B, and 6.9B. Our main results are shown in Fig. 1 which presents results for different configurations after 1 epoch of training (row 1) and including 4 uniform intermediate checkpoints (row 2). We include additional results on the training dynamics in Fig. 2, which shows win rates and KL bounds for intra-epoch training. We present our findings below.

**Model Over-Optimization:** We see clear over-optimization for all objectives as performance exhibits a hump-shaped pattern, where an additional increase in the KL budget leads to decreasing model performance. Moreover in Fig. 2 we observe similar intra-epoch training dynamics patterns as configurations with wider KL budgets achieve their best performance after training on only 25% of the data, after which performance starts decreasing in conjunction with increasing KL divergence metrics.

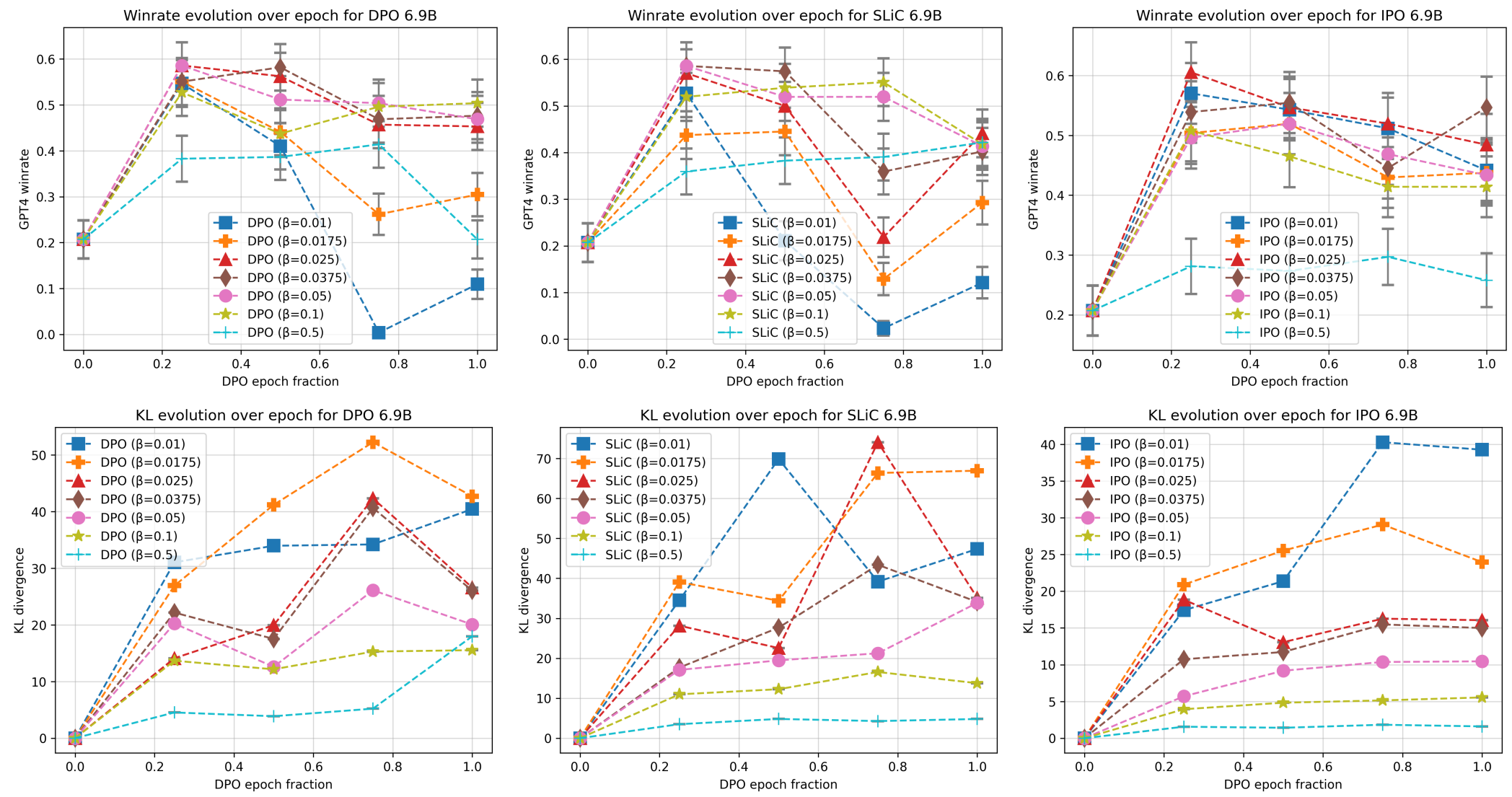

Figure 2: Results on intra-epoch optimization dynamics. The top row shows win-rates against the fraction of an epoch so far, while the bottom row shows the corresponding KL values. Under a lower KL constraint, most experiments reach their best performance in the first 25% of the epoch and degrade with additional training, while the model deviates from the reference under increasing KL. All models are 6.9B and vary across DPO, SLiC, and IPO loss formulations.

**Effect of Training Objective:** In the IPO work [4] the authors present theoretical arguments that due to the monotone sigmoid objective in the DPO formulation, the KL constraint is not effectively enforced and propose the quadratic fixed-margin loss as an alternative. Across all objectives, there are clear dependencies between the $\beta$ parameter and the corresponding KL achieved at the end of training. While DPO and SLiC exhibit similar performance, IPO indeed seems to be less prone to over-optimization and in general, achieve lower KLs under the same constraint. Our observations with IPO also align with prior works in preference-based RL and imitation learning where imposing a fixed margin led to more stable and performant methods [48, 51].

**Effect of Model Size:** The results also show a strong parameter count scaling effect. The Pythia 1B model achieves low performance under the same set of constraints it reaches much higher KL values, while almost immediately exhibiting signs of over-optimization. This behavior holds under all three objectives. At larger scales, the 6.9B Pythia model tends to exhibit more win-rate - KL trade-offs and be less prone to over-optimization, with both models significantly outperforming the 1B model. In the case of the IPO objective, the 6.9B also exhibits significantly better control over the KL objective and shows little to no over-optimization behavior.

### 3.2 Scaling Law Fits

Given we have established a framework for evaluating over-optimization in DAAs and empirically validated it (section 3.1), we now develop scaling laws for this phenomenon. Previous work in classical RLHF has established such scaling laws for reward model scores as a function of the KL divergence between initial and optimized policies [21]. The relevant functional of the reward $R(d)$ is

$$R(d) = d(\alpha - \beta \log d) \tag{5}$$

where $\alpha, \beta$ are constants dependent on the size of the reward model dataset and parameter count, and $d = \sqrt{D_{\text{KL}}(\pi||\pi_{\text{ref}})}$. As DAAs do not train a proxy reward model, we treat GPT4 winrates over dataset completions as a proxy for gold reward. Somewhat surprisingly, we find that this scaling law accurately relates $d$ and winrates for DAAs. Compared to a quadratic fit between $D_{\text{KL}}(\pi||\pi_{\text{ref}})$ and winrates, this scaling law halves the RMSE. It is worth noting, however, that a quadratic fit between $d$ and winrates yields a similar error compared to Equation 5.

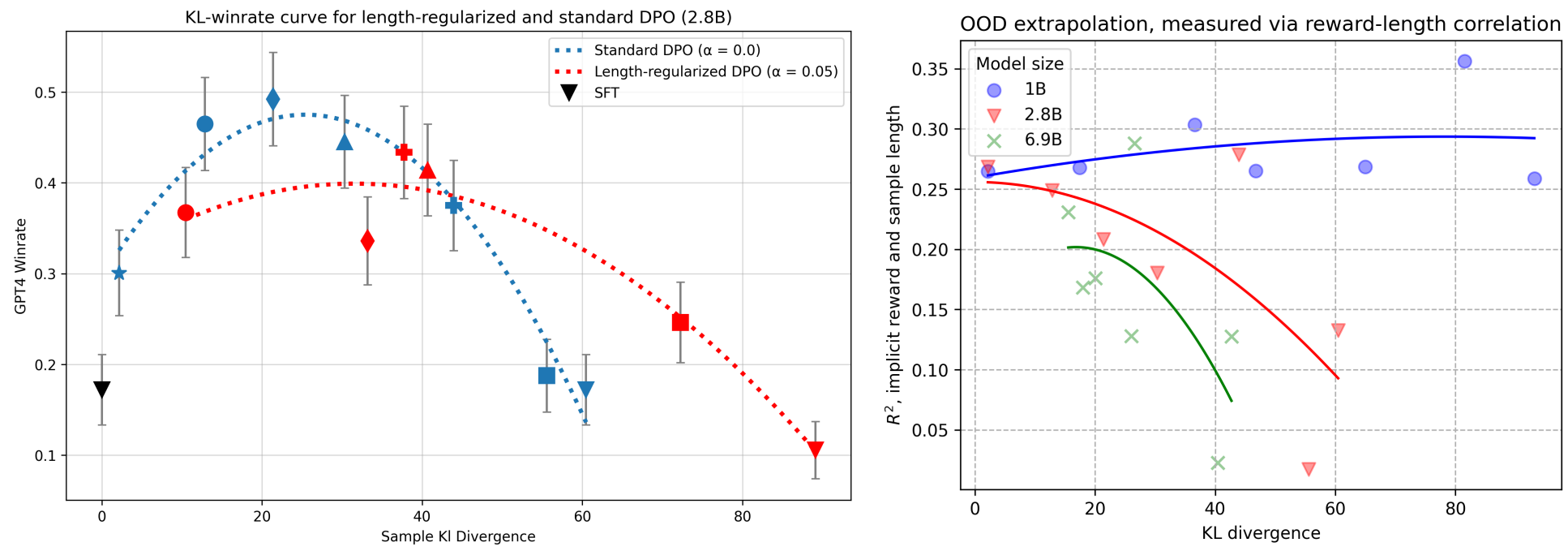

Figure 3: **Left:** KL budget versus win-rates (over dataset human answer) with and without length-regularization [44]. While including a length correction in the optimization objective changes the KL-win-rate Pareto Frontier, it does not alleviate reward over-optimization and might even exacerbate it. **Right:** Scaling behavior for length extrapolation - smaller capacity models (either by size or KL budget) extrapolate more strongly on simpler features such as length.

### 3.3 Length Correlations

Prior work [44] has shown that the DPO algorithm is prone to length exploitation as it amplifies verbosity biases in preference datasets. Here we show that length is not the only dimension on which exploitation can occur. Our experimental results are shown in Fig. 3. On the left, we show results for the 2.8B Pythia model with standard training plus the length-regularization approach from [44]. Both approaches suffer from over-optimization, but the dynamics differ depending on the KL budget. Moreover, even though the regularized model achieves higher win rates on a length-correct basis, it under-performs the model trained with the standard objective in the lower KL constraint region.

Recent work [27] has also shown that DAAs prioritize features of the data based on their complexity and prevalence (with length a clear example of human datasets). [44] further showed that models trained with the DPO algorithm extrapolate significantly based on length. We extend this analysis in Fig, 3 (right). We consider a linear regression of the form

$$\log \frac{\pi_\theta(y^{(i)}|x^{(i)})}{\pi_{ref}(y^{(i)}|x^{(i)})} = \hat{\gamma}|y^{(i)}| + \epsilon^{(i)} \tag{6}$$

where $x^{(i)}$ are held-out prompts and $y^{(i)}$ are samples from the corresponding model between the DPO implicit reward and length. We fit a different regression for each model size and checkpoint and plot the corresponding $R^2$ values. We observe two main effects; first, there is a clear scaling law behavior. Weaker models extrapolate across the simple length feature to a much higher degree than stronger ones. This is especially clear when comparing the behavior of the Pythia 1B versus the 2.8B and 6.9B models. Second, we see significant effects based on the KL budget - under a smaller budget all model sizes exhibit higher extrapolation behavior. Based on these results we formulate the hypothesis that under limited capacity, either from model capability or limited KL budgets, the model will extrapolate more strongly based on simpler features, which can lead to OOD issues.

### 3.4 Reward Metrics Correlations

Prior works have measured reward model quality in ranking settings by classification accuracy. We evaluate the relationship between the DAA implicit reward model accuracy and policy performance in Figure 4. The DPO and SLiC algorithms exhibit little to no correlation between reward model accuracy and downstream model performance. The IPO model shows a weak positive relationship, but upon further examination, this is entirely due to model size scaling - stronger models both fit the data better and produce better generations as well, however within each particular model size, there is no discernible relationship between the DAA implicit reward accuracy and the actual policy performance. Similar observations hold when comparing the empirical DAA loss with model performance, which is contrary to observations in supervised pre-training and instruction tuning [30].

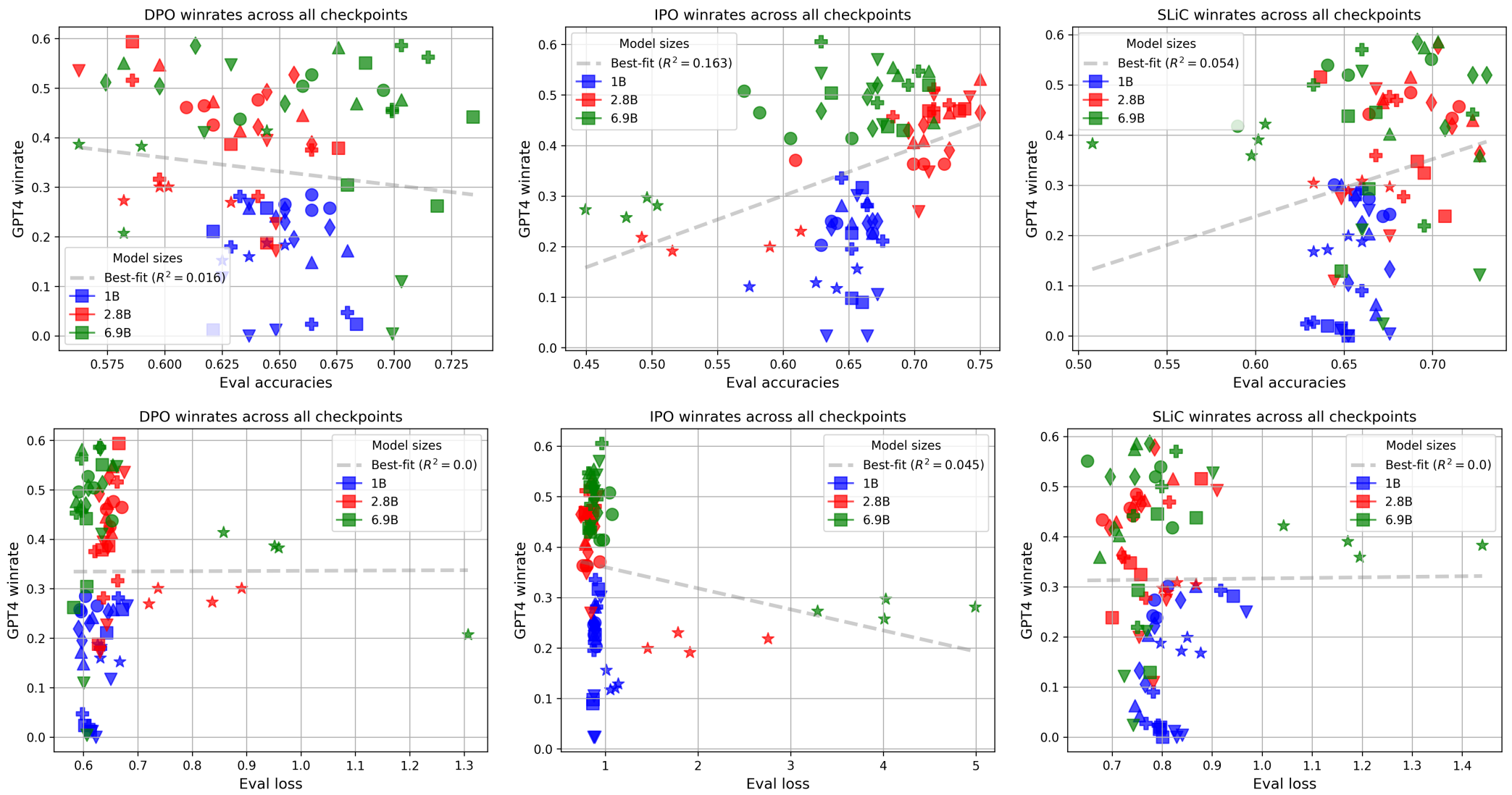

Figure 4: **Top:** We plot the DAA implicit reward accuracy in preference classification versus win rates. **Bottom:** DAA optimization loss versus checkpoint win rates. Model training statistics, do not exhibit a strong relationship with downstream performance.

### 3.5 Decreasing Likelihoods and Model Performance

A number of recent works have observed that the implicit DAA rewards of both preferred and dis-preferred responses decrease during training, which may be counter-intuitive. In [47] the authors make a counter-point that in offline training of DAAs $\pi_{\text{ref}}$ is usually pre-trained with SFT on the preferred response and thus

$$\mathbb{E}_{p_{\mathcal{D}}(y_w|x)}\left[\log \frac{\pi_\theta(y_w|x)}{\pi_{\text{ref}}(y_w|x)}\right] \approx \mathbb{E}_{\pi_{\text{ref}}(y_w|x)}\left[\log \frac{\pi_\theta(y_w|x)}{\pi_{\text{ref}}(y_w|x)}\right] = -\mathbb{D}_{\text{KL}}\left[\pi_{\text{ref}}(y|x) \mid\mid \pi_\theta(y \mid x)\right] \quad (7)$$

where $p_{\mathcal{D}}(y^w|x)$ is the dataset distribution of preferred answers. That is the expected implicit reward represents a forward KL divergence between the reference policy and the optimization policy, thus it is expected to be negative and decrease with training as the optimization model moves away from the reference. In this section, we study whether this empirical phenomenon presents a challenge for DAA learning. Similar to Fig. 1 we plot the win rates against the square-root-transformed (negative) expected implicit reward of the preferred response (evaluated on a held-out evaluation dataset), which as stated above approximates the (square-root-transformed) forward KL $\mathbb{D}_{\text{KL}}\left[\pi_{\text{ref}}(y|x) \mid\mid \pi_\theta(y \mid x)\right.$. Results are included in Fig. 5, which follow closely the pattern in Fig. 1 with performance initially increasing before it starts dipping down after a certain threshold. This indicates that under the standard DAA training pipeline decreasing likelihoods are not necessarily an issue for performance, and are even necessary for improvement, but exhibit non-linear over-optimization dynamics.

## 4 Reward Exploitation in Direct Alignment Algorithms

While the phenomena observed in the previous section echo those observed in classical RLHF, their underlying causes may be distinct. Reward over-optimization in classical RLHF is largely attributed to querying a proxy reward function that is potentially OOD, while DAAs do not train a separate reward model. Instead, DAAs are generally understood as fitting an "implicit" reward model to preference data with the parameterization $r_\theta(x, y) = \beta \log \frac{\pi_\theta(y|x)}{\pi_{\text{ref}}(y|x)}$ using the objective in eq. (2). Therefore, the OOD behavior of the policy is inextricably linked to the OOD behavior of the implicit reward model. We demonstrate below that the reward modeling objective used is heavily under-constrained, allowing for a potentially large number of solutions that can place weight on OOD responses. This is especially problematic for DAAs which deterministically map the optimal policy from the "implicit" reward.

**Rank Deficiency with Finite Preferences.** In DAAs, the language modeling problem is treated as a contextual bandit. However, the space of possible prompts $x \in \mathcal{X}$ and answers $y \in \mathcal{Y}$ are both

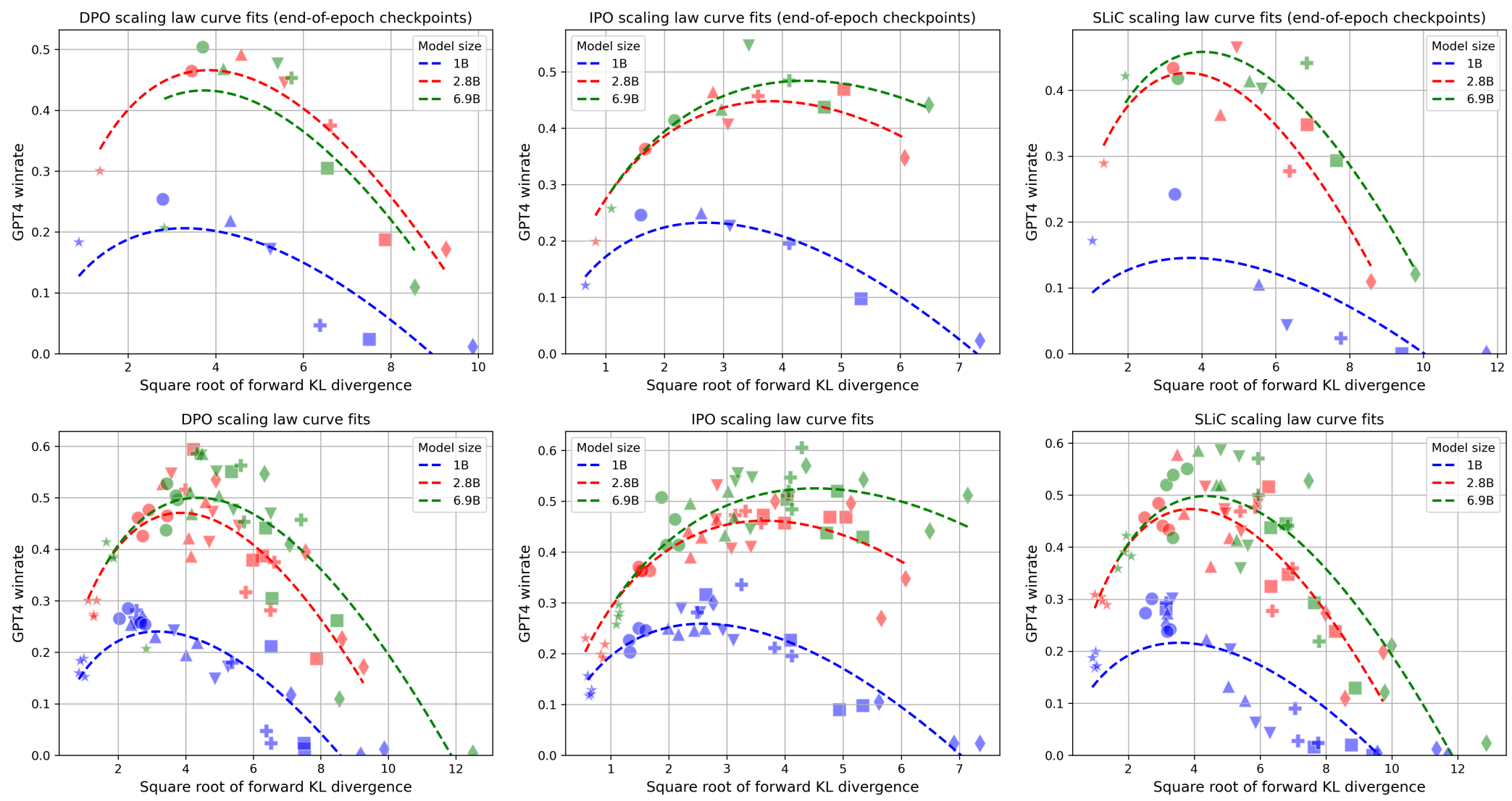

Figure 5: Over-optimization results for $\sqrt{\text{Forward KL}}$ vs. winrates. The top row shows the final performance after 1 epoch of training, while the second row also includes 4 intermediate checkpoints. The fitted dotted curves are scaling laws from [21] applied to DAAs, with GPT4 winrates taking the place of the gold reward model score.

exponentially large in sequence length. However, as highlighted by Tang et al. [59], DAAs often assume full support of the reference distribution when mapping from the implicit reward to the optimal policy $\pi$ by eq. (10). However, in practice such coverage is impossible. Instead, preference datasets cover a minuscule portion of the prompt-response space. Unfortunately, as DAA objectives are not strictly convex, their loss functions (eq. (4)) can have multiple global optimas, which may be undesirable. We demonstrate this below, using the regression interpretation from Hejna et al. [23].

First, we re-write the DAA objective from eq. (4) using vectors in the prompt-response space $\mathcal{X} \times \mathcal{Y}$. Each preference query in the comparison dataset can be written as difference between indicator vectors, specifically $q_i = \mathbb{1}\{(x, y) = (x^{(i)}, y_w^{(i)})\} - \mathbb{1}\{(x, y) = (x^{(i)}, y_l^{(i)})\}$. This "query" vector simply selects the comparison from the prompt response space, with the entree corresponding to $(x, y^w)$ being +1 and the entree corresponding to $(x, y^l)$ being -1. Similarly, we can consider the learned policy to be a vector $\log \pi - \log \pi_{\text{ref}} \in \mathcal{X} \times \mathcal{Y}$, to which the distributional constraint also applies in practice. Our generalized DAA loss function can then be re-written as

$$\mathcal{L}_{\text{DAA}}(\pi_\theta, \mathcal{D}) = \sum_{i=1}^{|\mathcal{D}|} g\left(\beta q_i^\top \left(\log \pi(y|x) - \log \pi_{\text{ref}}(y|x)\right)\right), \text{ where } q_i[x, y] = \begin{cases} 1 & \text{if } (x, y) = (x^{(i)}, y_w^{(i)}) \\ -1 & \text{if } (x, y) = (x^{(i)}, y_l^{(i)}) \\ 0 & \text{otherwise} \end{cases}$$

with finite data. Choosing $g$ to be the negative log sigmoid above recovers DPO with finite preferences, but also logistic regression with a data matrix $Q$ of shape $|\mathcal{D}|$ by $|\mathcal{X} \times \mathcal{Y}|$ constructed by stacking the aforementioned query vectors $q$. As $|\mathcal{X} \times \mathcal{Y}| >> |\mathcal{D}|$, this matrix is likely to have a non-trivial null space, making the problem not strictly convex. Thus, there are many possible policies $\pi$ that can achieve the same optima, some of which will place a high weight on out-of-distribution responses due to the distributional constraint of policy [23, 70]. To demonstrate this, we formalize the construction below.

**Proposition 1** *(Adapted from Hejna et al. [23]) Let $S$ be the set of win-or-lose prompt-response vectors $(x, y)$ in $\mathcal{D}$. Provided:*

1. *The intersection of the null space of $Q$, $N(Q)$, and the span of $S$, span$(S)$, is non-trivial.*
2. *For every $x$ there exists a response $y_{OOD} \in \mathcal{Y}$ that is not in the data, $(x, y_{OOD}) \neq S$.*

*Then, there are infinite number of minima to eq. (4) which place weight on out-of-distribution responses $y$.*

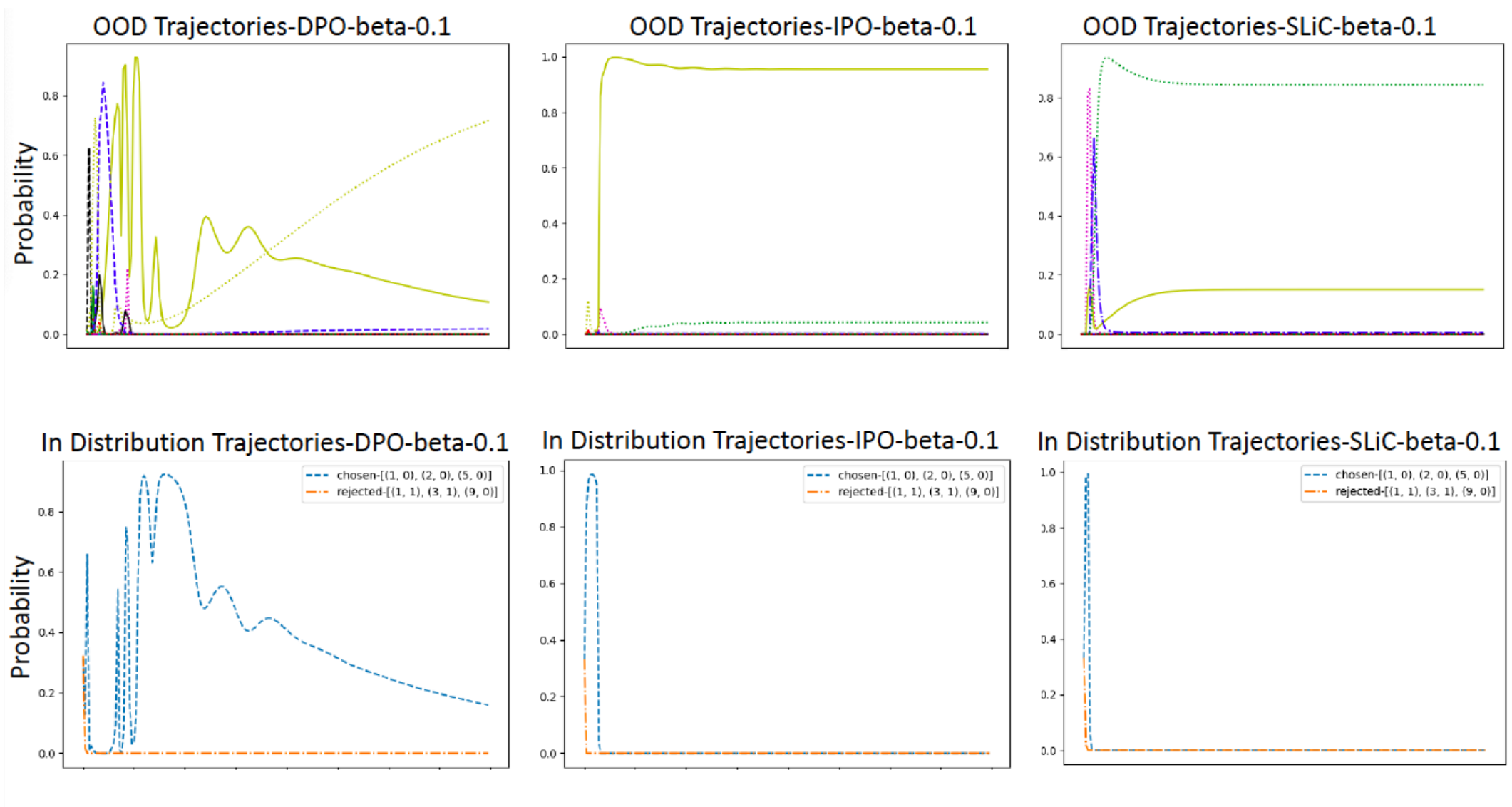

Figure 7: (Top row) Probability of OOD trajectories. DAA algorithms end up placing a substantial probability mass of some of the OOD trajectories during training. (Bottom row) Probability of in-distribution (preference-pair) trajectories decreases during training.

*Proof.* Let $\hat{\pi}$ be the minima of the DAA loss function. Choose a vector $u$ such that $u \in N(Q)$, $u \in \text{span}(S)$, and $u$ has at least one negative component. Modifying the log policy vector as $\log \hat{\pi} + u$ will not affect the DAA loss, as $u$ is in the null space of $Q$, but the log-probability of the policy will decrease for least one prompt-response pair in $S$ by construction. However, $e^{\log \hat{\pi} + u}$ may not integrate to one. To fix this, we can construct a second vector $v \in N(Q)$ using the $y_{\text{OOD}}$ at each $x$ such that $e^{\log \hat{\pi} + u + v}$ integrates to one. For more details, we refer the reader to Hejna et al. [23] Appendix A.3.

The second constraint of proposition 1 is often trivially satisfied by the dimension of the response space as we are unlikely to see *every* response to a prompt. The first constraint is harder, but can be satisfied by conflicting preferences. A trivial example which satisfies these constraints is a simple MDP in which there is only a single state (or prompt $x$), but three possible actions (or responses), $y_1$, $y_2$, and $y_3$. If we construct the preference dataset $\mathcal{D} = \{(y_1 \succ y_2), (y_2 \succ y_1)\}$, omitting $x$ for brevity, then we satisfy the above conditions: the null space of $Q$ is non trivial in span of $y_1$ and $y_2$ and there is an out-of-distribution action $y_3$. In this setting, the DPO loss is minimized by both $\hat{\pi}(y|x) = (0.5, 0.5, 0)$ and $\hat{\pi}(y|x) = (0.0, 0.0, 1.0)$. In fact, it is minimized by infinitely many policies which place equal weight on $y_1$ and $y_2$. To demonstrate this effect in higher dimensions across a number of different DAA methods, we conduct experiments in a Toy MDP which bears resemblance to the language modeling setting.

**Understanding OOD behavior for DAA algorithms with a Toy MDP**: To illustrate that DAA algorithms, in general and not an artifact of training LLM's, end up placing probability mass on OOD sequences during training we design a simple Tree MDP (shown in Figure 6) to mimic the token-level MDP in LLMs. We use a dataset containing a single preference between two trajectories and follow the standard procedure of running SFT on preferred responses before updating an RNN policy using a DAA. Figure 7 shows that even in this simple setup, popular DAAs (DPO/IPO/SLiC) end up extrapolating incorrectly out of distribution revealing a fundamental shortcoming. Unlike in standard RLHF, the non-strict convexity of the reward function in DAAs ends up directly affecting the policy. Detailed experimental details can be found in Appendix E.

## 5 Related Work

Broadly, over-optimization has been a widely studied phenomenon across different settings [60, 18]. Over-fitting can be characterized as over-optimization in the supervised learning setting [39, 32], which can harm generalization [19, 12, 24] or lead to susceptibility to adversarial attacks [56, 37, 15]. Reward hacking in reinforcement learning (RL) [54], where an agent maximizes its reward through

behavior that deviates from the intended goal, can be viewed as a different type of over-optimization, commonly observed in prior work [43, 3, 22].

We study over-optimization in the context of aligning LLMs with human feedback, for which the most common approach is RLHF as outlined in section 2.1. Similar RLHF techniques were originally pioneered for control [31, 2, 10]. Standard RLHF methods suffer from both potential over-fitting of the reward function and reward exploitation by the RL algorithm. Several works have considered how to reduce over-fitting or increase the robustness of learned reward functions using ensembles [13, 67, 16] or data smoothing [70]. Other approaches, like Moskovitz et al. [40] consider how reward exploitation can be reduced by using different optimization techniques in the RL stage. Much of this work is motivated by Gao et al. [21], which first characterized and provided scaling laws for over-optimization in RLHF.

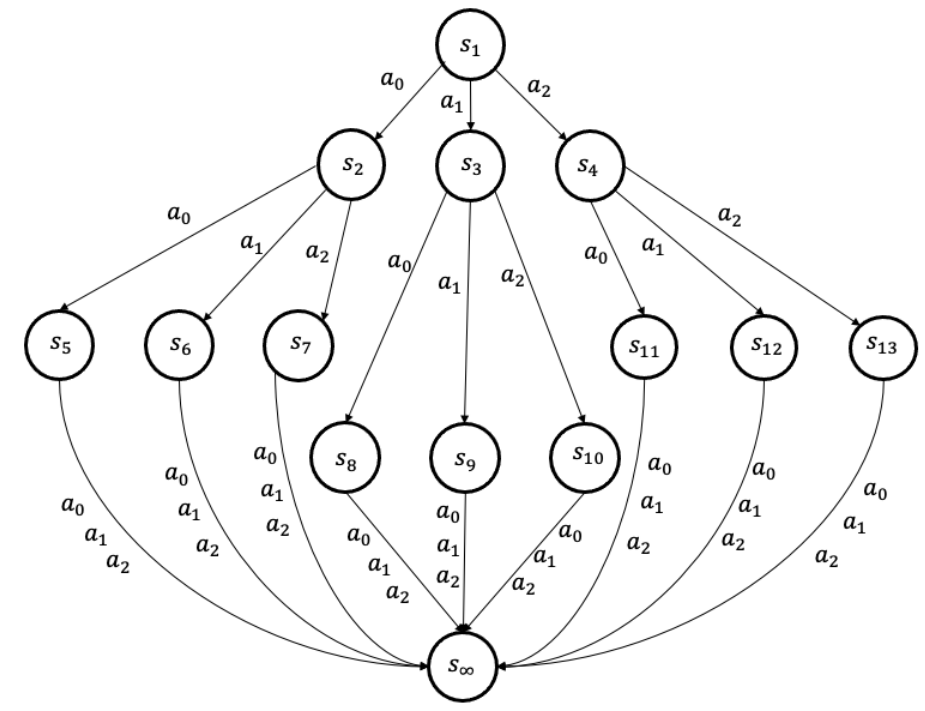

Figure 6: An illustration of the Tree MDP. At each state, we can choose one of 3 actions $(a_0, a_1, a_2)$, which deterministically maps to the next state. Furthermore, all the leaf nodes in this tree MDP, transition to the terminal absorbing state $s_\infty$, irrespective of the chosen action

Unlike Gao et al. [21], we consider the over-optimization problem in DAAs, which differs significantly from the standard RLHF pipeline. Different DAAs have been derived theoretically [47, 46, 68, 4, 64], and applied to problems beyond language modeling like image generation [63] and control [23]. In all of these scenarios, over-optimization problems have persisted. Park et al. [44] show that DAAs commonly over-fit to length and the expense of performance, which has been linked to inherent bias in training data [53, 29]. Other works have tried to allow DAAs to use more types of data like demonstrations [49] or ratings [17] to get better performance. Recently, incorporating online data has proven critical to improving performance [66, 26, 57]. Concurrent to our work, Tang et al. [58] study the differences between offline DAAs and standard RLHF methods. Unlike us, they focus on comparisons with online sampling whereas we focus on the purely offline setting.

## 6 Conclusion

In this work, we present an analysis of the over-optimization problem in Direct Alignment Algorithms. Through extensive experimentation on different algorithms (DPO, IPO, SLIC) and at different model scales (1B, 2.8B, 6.9B), we observe consistent over-optimization trends at different KL-divergence budgets. While our analysis is a first step, it is not a complete picture of understanding the over-optimization phenomena. More work can be done characterizing this effect at larger model scales, which we were unable to do due to computational limitations. Nevertheless, we believe our work sheds light on important problems in Direct Alignment Algorithms that can spur future research.

### Acknowledgments

This work has taken place in part in the Safe, Correct, and Aligned Learning and Robotics Lab (SCALAR) at The University of Massachusetts Amherst. SCALAR research is supported in part by the NSF (IIS-2323384), AFOSR (FA9550-20-1-0077), and the Center for AI Safety (CAIS). This work has taken place in part in the Rewarding Lab at UT Austin. The Rewarding Lab is supported by NSF (IIS-2402650), ONR (N00014-22-1-2204), EA Ventures, Bosch, UT Austin's Good Systems grand challenge, and Open Philanthropy.

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

## A Limitations and Societal Impacts

Our discussion highlights a number of issues with direct alignment algorithms used widely as means to align to human values. This work has mostly focused on pointing out those issues along with a theoretical underpinning of the issue but does not provide a way to resolve these issues. We still assume an underlying model of human preferences, which is an ongoing research area as no model is perfect in explaining the ways humans give preferences. Our work aims to drive the push towards better alignment algorithms that do not overoptimize and generate models that are safe to be deployed in our society. We believe only through understanding and demonstrating the shortcomings of current methods we can develop better alignment methods.

## B Experiment Details

We largely follow the DPO setup unless otherwise mentioned and build on their code (https://github.com/eric-mitchell/direct-preference-optimization) without changing any hyperparameters unless otherwise mentioned.

For all DAA experiments, we used the curated OpenAI TL;DR dataset with 92K preferred-dispreferred summary completions [55]. Each prompt is a Reddit post belonging to one of several topic forums, with title/post metadata included. $256$ prompts sampled from the held-out set are used for all evaluations (e.g. loss, accuracy, KL, winrates, length), with temperature $1.0$ and max length $512$.

Model sizes include 1B, 2.8B, and 6.9B and were initialized from the base Pythia pre-trained weights. All models underwent supervised fine-tuning on TL;DR prior to direct alignment. Across all SFT and DAA runs, we used a batch size of 128 (8 gradient accumulation steps), and RMSProp with a learning rate of $0.5 \times 10^{-6}$ (linear warmup for 150 steps) for 1 epoch. 1B models were trained on 2 NVIDIA A40 GPUs, 2.8B models were trained on 4 NVIDIA A40 GPUs, and 6.9B models were trained on 4 NVIDIA A100 GPUs. All evaluations were computed with "gpt-4-turbo-2024-04-09" as judge, with random positional flips to avoid known bias.

## C Appendix A: Complete Intra-Epoch Training Dynamics

This appendix contains similar intra-epoch KL divergence and winrate evolution results as in Fig. 2, across all model sizes.

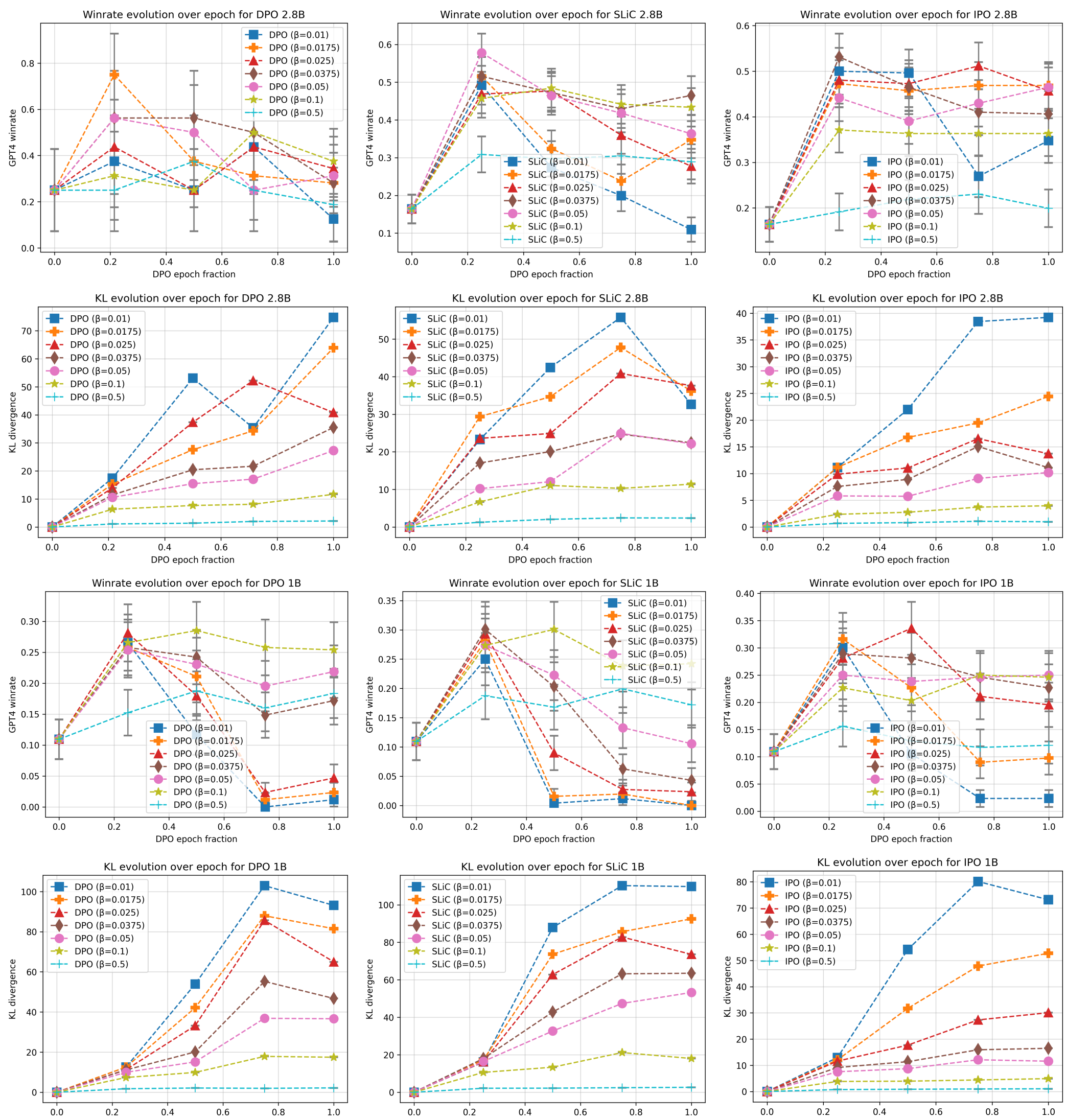

Figure 8: KL divergence and GPT4 winrate evolution for 2.8B and 1B models across DPO, SLiC, and IPO losses. Similar to the 6.9B models, performance tends to degrade after the first quarter epoch, particularly under a low KL budget, while KL increases almost monotonically.

## D Overoptimization from the lens of Implicit Bootstrapping

Reward over-optimization is well understood in the classical RLHF setting, with a consensus that is driven by two main components - using a proxy reward function that is trained on limited data and continuous querying with new, potentially OOD samples during PPO training. At first glance, none of these conditions hold in DAAs as we do not train a separate proxy reward model or generate new data during training. Therefore, understanding reward over-optimization in DAAs requires a new theory. We will base our analysis on [47] using the token-level MDP and corresponding (soft) Q-learning formulation. Consider the class of dense per-token reward functions $r_\theta(x, y_{\leq i})$, where $y_{\leq i}$ denotes the first $i$ tokens of $y$, with sequence level-reward $r_\theta(x, y) = \sum_{i=1}^{|y|} r_\theta(x, y_{\leq i})$. This is a strictly more general class than the sparse reward function which returns a single score at the end of the sequence since we can set all intermediate rewards as 0. Within the framework of [47] given a DAA-trained policy $\pi_\theta$, there exists a dense per-token reward $r_\theta$, that minimizes the reward modeling objective in Eq. 2 and satisfy the below.

The (soft) Bellman Equation holds:

$$Q^*(y_i, (x, y_{<i})) = \begin{cases} r(x, y_{\leq i}) + \beta \log \pi_{\text{ref}}(y_i|(x, y_{<i})) + V^*((x, y_{\leq i})), & \text{if } y_i \text{ is not } \textbf{EOS} \\ r(x, y_{\leq i}) + \beta \log \pi_{\text{ref}}(y_i|(x, y_{<i})), & \text{if } y_i \text{ is } \textbf{EOS} \end{cases} \tag{8}$$

where $V^*$ is the corresponding soft-value function:

$$V^*((x, y_{<i})) = \beta \log \sum_{y \in |V|} e^{Q^*(y,(x,y_{<i}))/\beta} \tag{9}$$

then the DAA policy $\pi_\theta$ satisfies:

$$\pi_\theta(y_i|(x, y_{<i})) = \exp(\frac{1}{\beta} Q^*(y_i, (x, y_{<i})) - V^*((x, y_{<i}))) \tag{10}$$

in this interpretation, the LLM logits $l_\theta[i] = Q^*(y_i, (x, y_{<i}))/\beta$ represent Q-values. With a direct substitution, we then have

$$Q^*(y_i, (x, y_{<i})) = r(x, y_{\leq i}) + \beta \log \pi_{\text{ref}}(y_i|(x, y_{<i})) + \underbrace{\beta \log \sum_{y_i \in |V|} e^{Q^*(y,(x,y_{<i}))/\beta}}_{\text{OOD bootstrapping}} \tag{11}$$

That is in this framework DAAs may suffer from the classical OOD bootstrapping issue in offline RL [20, 35, 33, 52]. In this case, even though the objective is trained fully offline we still effectively query the model on the values of unseen tokens. This interpretation also provides further insight into the effect of the $\beta$ coefficient and the training dynamics. For small values of beta the estimate

$$\beta \log \sum_{y_i \in |V|} e^{Q^*(y,(x,y_{<i}))/\beta} \approx \max_{y \in |V|} Q^*(y, (x, y_{<i})) \tag{12}$$

that is smaller parameter values yield a more optimistic estimate, which results in a higher level of OOD bootstrapping. This interpretation would also explain the somewhat counter-intuitive results of section 3.4. While the implicit reward function can adequately fit and model the data, the resulting LLM might behave sub-optimally, due to OOD bootstrapping in the corresponding Q-value estimate.

## E Understanding Behavior of DAAs on OOD sequences

We have established that common DAA objectives allow for placing a high likelihood on OOD data. In practice, while one might expect the likelihood of preferred responses to increase during training, it has been observed that algorithms like DPO decrease the likelihood of both the preferred and dis-preferred responses [42]. In fact, this is expected from a max-entropy RL perspective [47]. Since the total probability mass must sum to one, the probability of OOD responses must increase during the course of training. A small amount of extrapolation may be necessary to reach the optimal policy, however, too much is potentially detrimental to performance. Because they are not adequately constrained to the reference distribution, current DAA objectives allow this to happen.

To understand how DAAs allocate probability mass out of distribution, we use a toy Markov Decision Process (MDP), that mimics the LLM setting. The MDP is modeled as a tree, originating from a single start state, featuring deterministic transitions. The Toy MDP is illustrated in fig. 6.

### E.1 Designing a toy LLM MDP

The MDP is modeled as a tree, originating from a single start state. This configuration mirrors the token-level MDP in Direct Preference Optimization (DPO) [47], or the scenario where both preferred and dispreferred responses are conditioned on the same prompt in the broader Large Language Model alignment context. Each leaf node in the MDP transitions deterministically to a terminal absorbing state, regardless of the action taken. The deterministic transitions resemble the LLM setting, where the current state is represented by the sequence of encountered tokens $(s_1, s_2, ..., s_i)$, and the action corresponds to predicting the next word $s_{i+1}$ from the vocabulary, given the context. In this simplified MDP, the deterministic transition is akin to a concatenation function, advancing the state to the next step $(s_1, s_2, ..., s_i, s_{i+1})$. Employing a toy MDP enables us to systematically evaluate the trajectory probabilities for all feasible paths within the MDP, shedding light on the allocation of probability mass by Direct Alignment Algorithms (DAAs) towards out-of-distribution (OOD) trajectories.

**The Experimental Setup.** We adhere to the standard direct alignment protocol [46][41], encompassing two key stages:

1. **Supervised Fine-tuning (SFT) / Behavioral Cloning (BC):** This phase involves fine-tuning the policy based on a limited number of trajectories. Specifically, we utilize three demonstrations for SFT: $(s_1, a_0, s_2, a_0, s_5, a_0, s_\infty)$, $(s_1, a_1, s_3, a_1, s_9, a_0, s_\infty)$, and $(s_1, a_2, s_4, a_2, s_{13}, a_2, s_\infty)$.
2. **Alignment with Preferences:** In this stage, preferences extracted from trajectories are employed to align the policy. Notably, we have only one preference available: $(s_1, a_1, s_3, a_1, s_9, a_0, s_\infty) \succ (s_1, a_0, s_2, a_0, s_5, a_0, s_\infty)$. This deliberate constraint exaggerates a scenario with limited data, enabling us to gauge the probability mass allocated to out-of-distribution (OOD) trajectories under such conditions. Insights garnered from this exaggerated low-data scenario hold relevance for Large Language Model (LLM) settings where preference datasets used for alignment are notably smaller compared to the scale of LLM models deployed.

We utilize a Recurrent Neural Network (RNN) policy to navigate through the MDP, facilitating a closer resemblance to real-world language modeling scenarios.

Subsequently, we explore three distinct direct alignment loss functions: Direct Preference Optimization (DPO) [46], Identity Preference Optimization (IPO) [4], and Sequence Likelihood Calibration (SLiC) [68]. Additionally, we investigate how the selection of the KL penalty coefficient $\beta$ influences the distribution of probability mass on OOD trajectories. This exploration encompasses three values of $\beta$: $(0.01, 0.1, 0.5)$.

In general, the plots illustrate that Direct Alignment Algorithms (DAAs) tend to allocate a significant proportion of the probability mass to out-of-distribution (OOD) trajectories during the alignment process. While Figure 9 may suggest that Direct Preference Optimization (DPO) can retain a substantial amount of probability mass on the selected trajectory in the preference dataset, it's noteworthy that the plots for DPO exhibit considerable noise. To provide further insight, Figure 18 displays the plots resulting from three additional repetitions of the DPO experiment. Similar noisy trends were also observed in the experiments for IPO and SLiC. This elucidates the unconstrained

nature of the DPO problem: multiple solutions exist for the DPO loss, each distributing varying amounts of probability mass to OOD trajectories. In the experiments with IPO and SLiC, it's also observed that similar to DPO, the probability mass allocated to in-distribution trajectories can diminish substantially over the course of training. Notably, the probability mass, in our experiments, becomes concentrated on a select few out-of-distribution trajectories. Moreover, consistent trends are discernible across various values of $\beta$. The results of our experiments with the Toy-MDP can be found in the following figures 12, 9, 15, 13, 10, 16, 14, 11, 17.

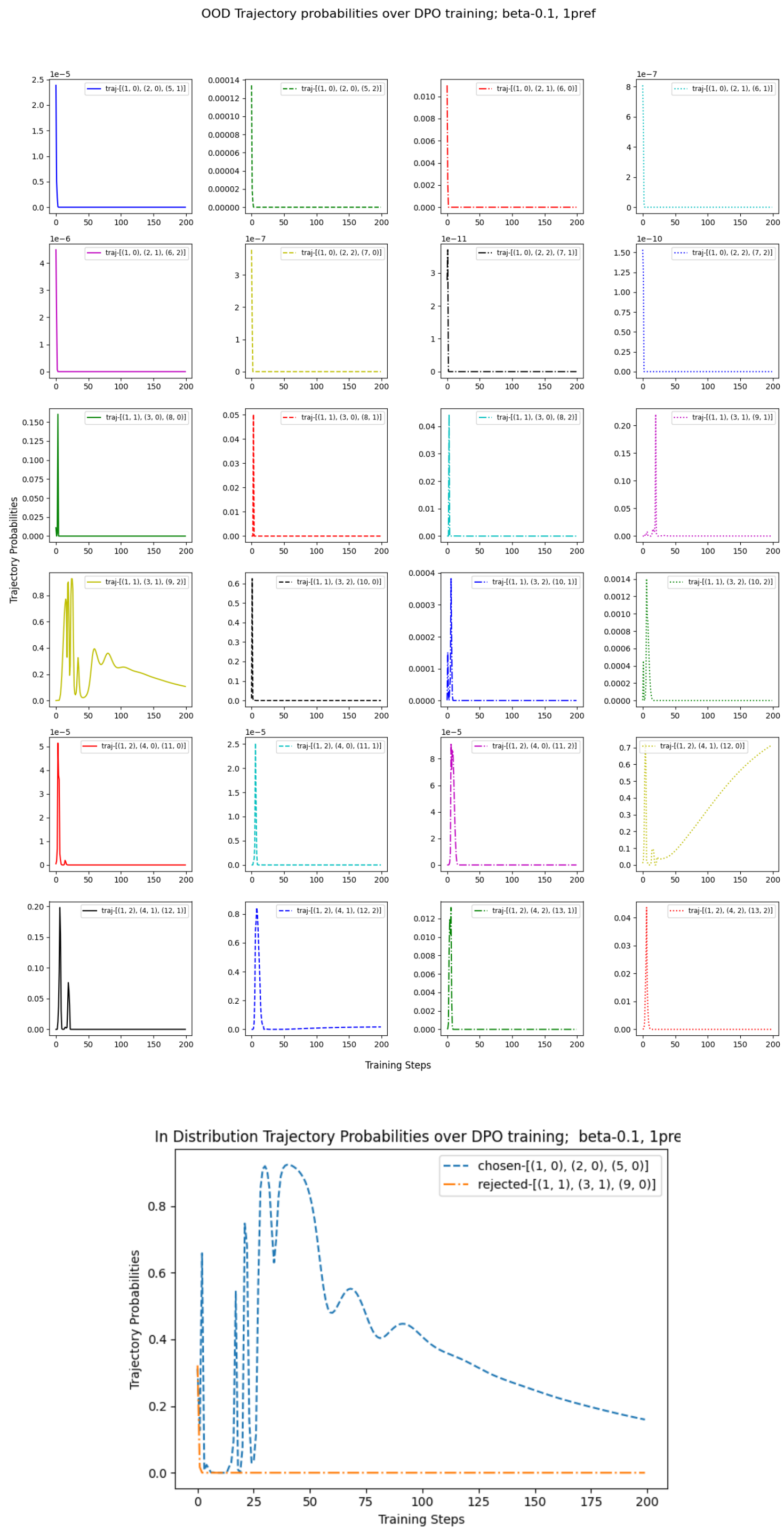

Figure 9: Trajectory probabilities throughout DPO training, $\beta = 0.1$. The top plot shows how the probability mass of different OOD trajectories, changes throughout training. The bottom plot shows how the probability mass of the trajectories in our preference dataset (size 1) changes over training. The trajectories are listed in the legends for the plots, as a sequence of state, action pairs.

.

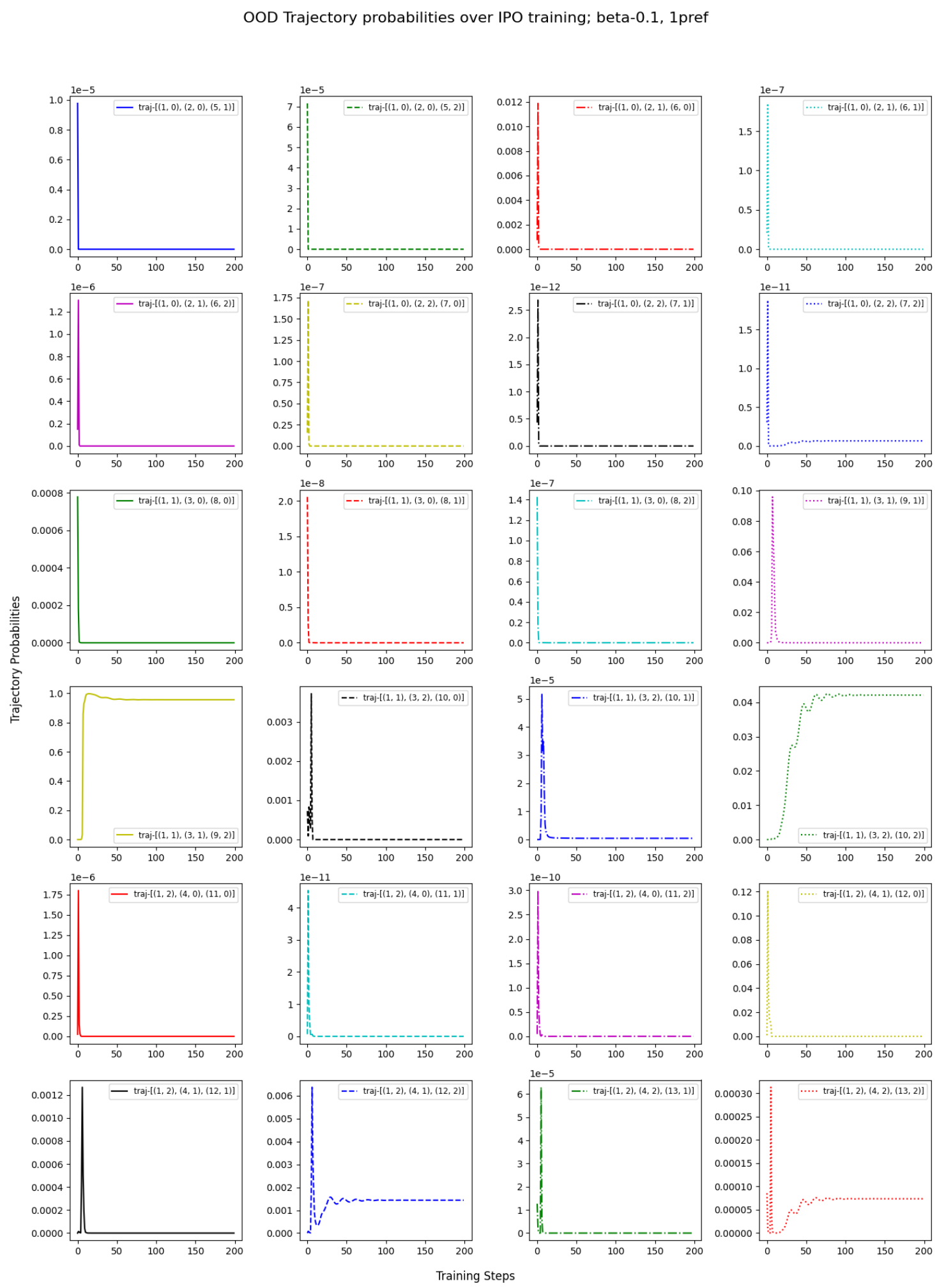

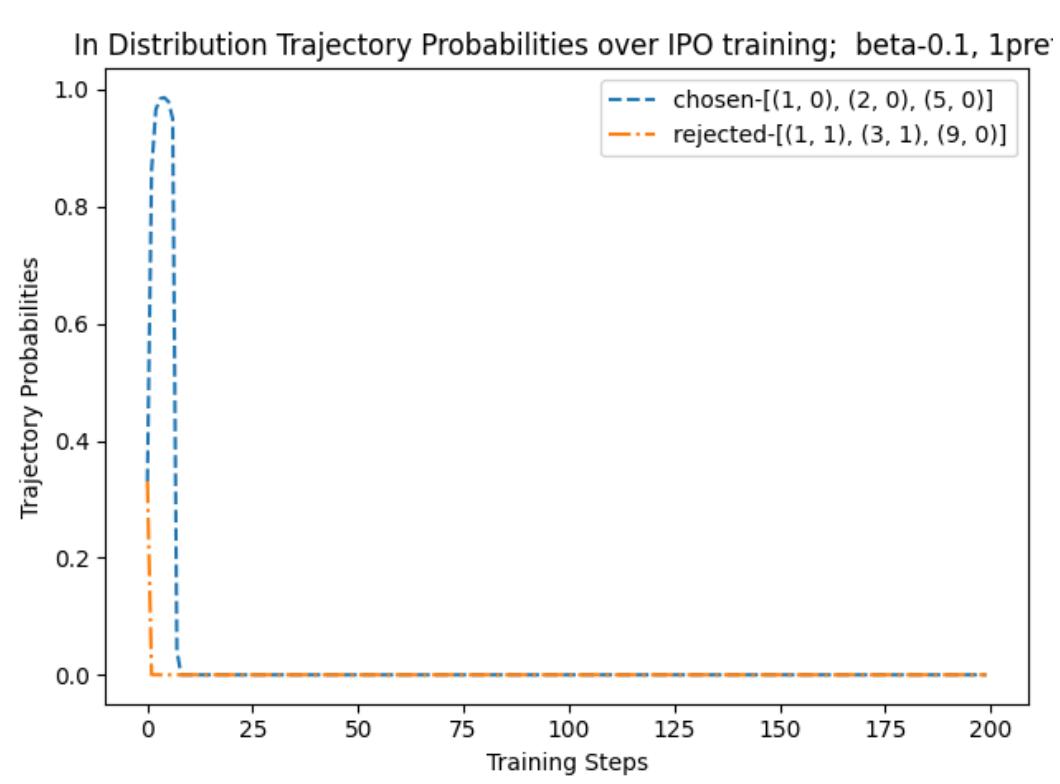

Figure 10: Trajectory probabilities throughout IPO training, $\beta = 0.1$. The top plot shows how the probability mass of different OOD trajectories, changes throughout training. The bottom plot shows how the probability mass of the trajectories in our preference dataset (size 1) changes over training. The trajectories are listed in the legends for the plots, as a sequence of state, action pairs.

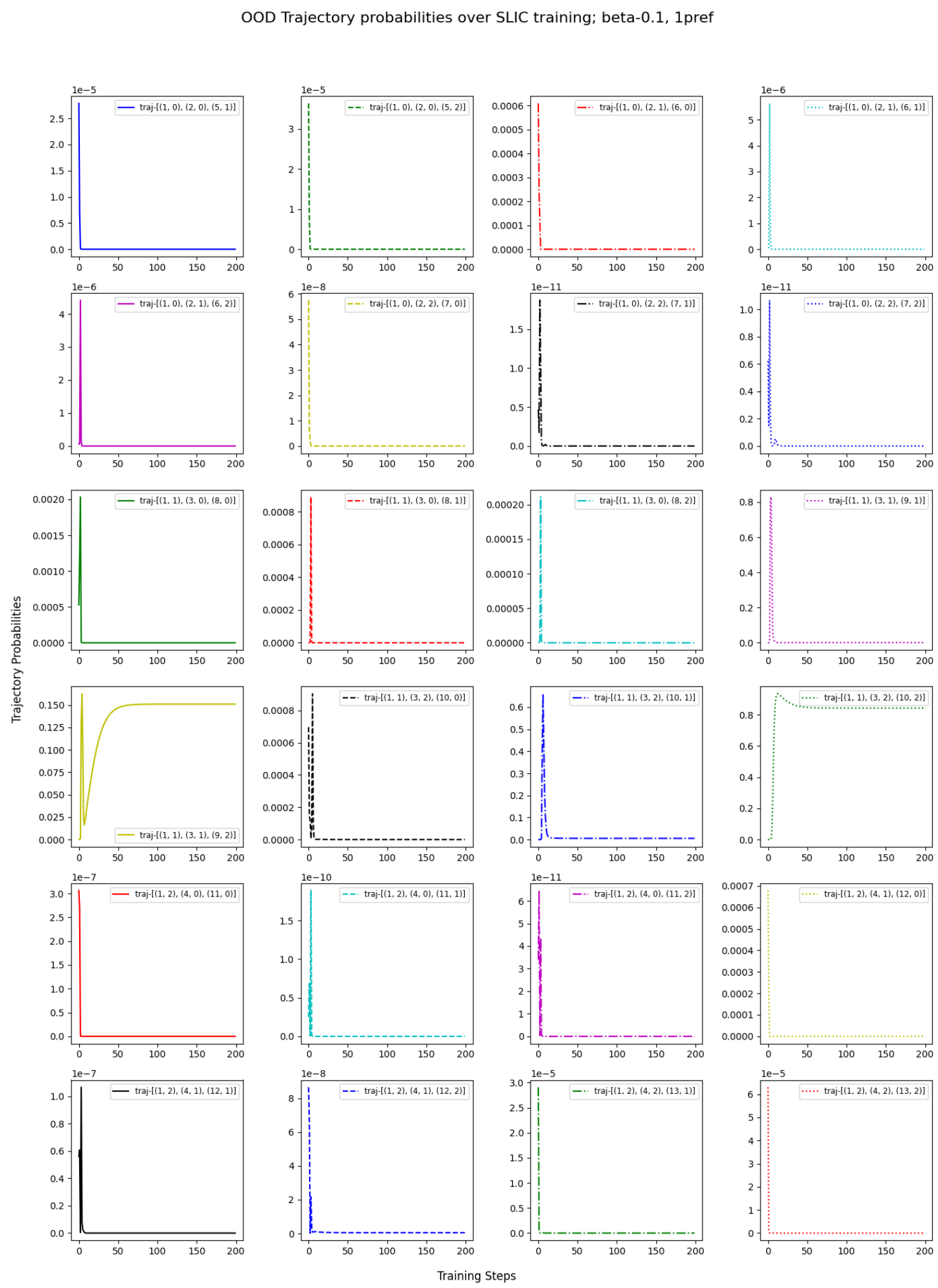

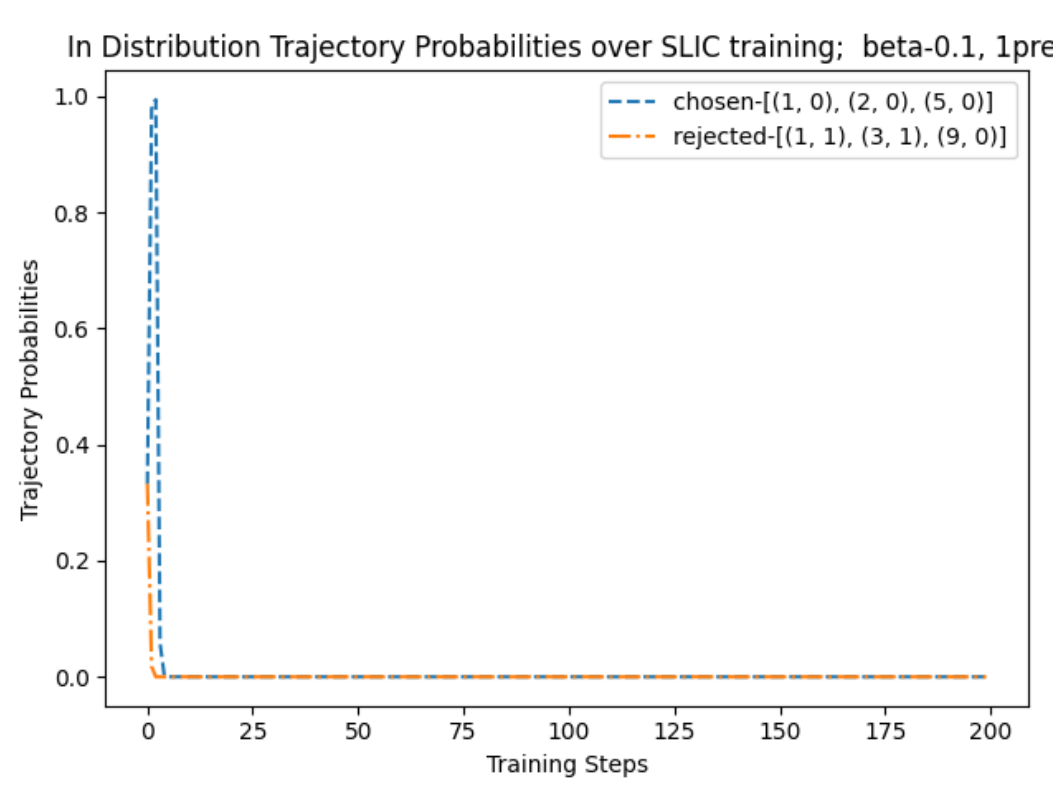

Figure 11: Trajectory probabilities throughout SLiC training, $\beta = 0.1$. The top plot shows how the probability mass of different OOD trajectories, changes throughout training. The bottom plot shows how the probability mass of the trajectories in our preference dataset (size 1) changes over training. The trajectories are listed in the legends for the plots, as a sequence of state, action pairs.

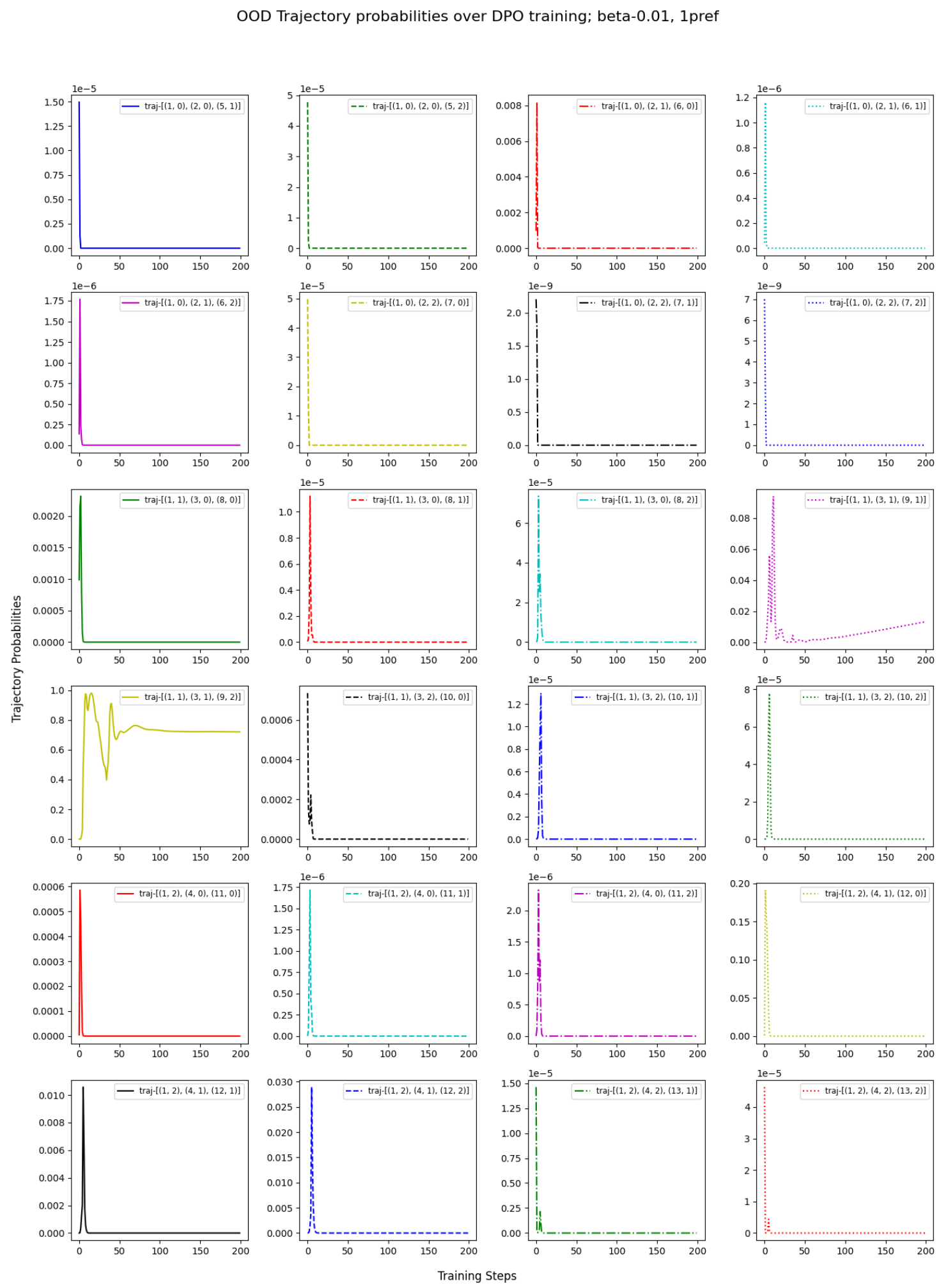

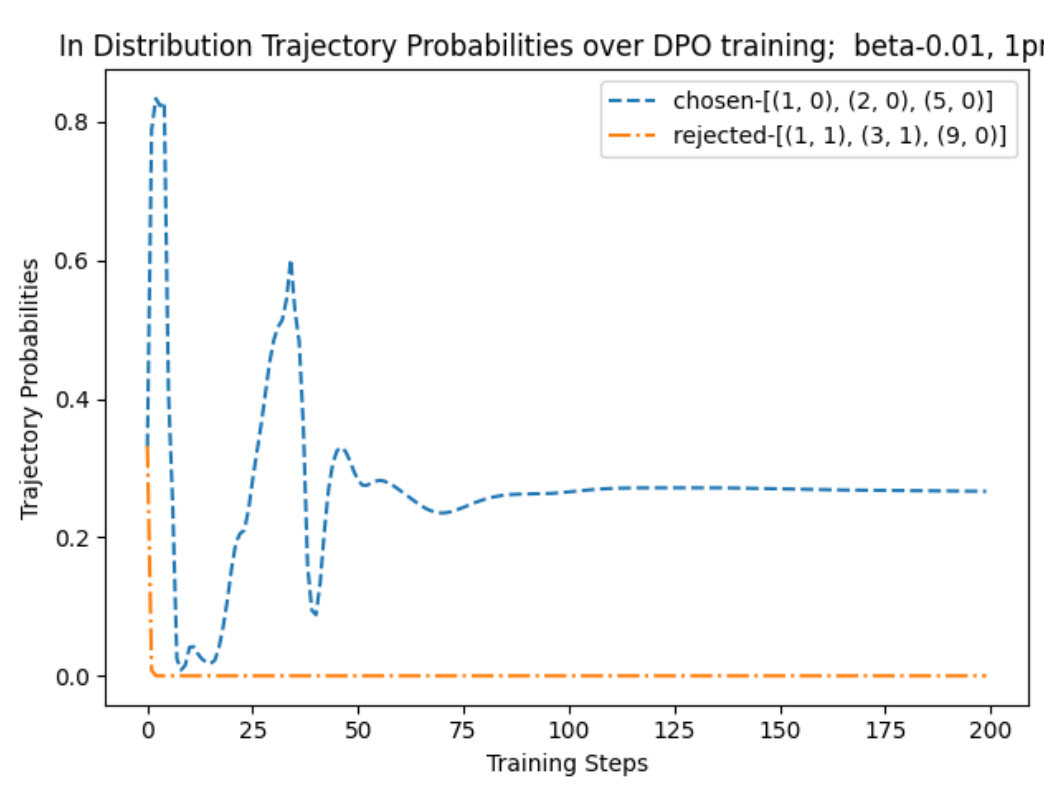

Figure 12: Trajectory probabilities throughout DPO training, $\beta = 0.01$. The top plot shows how the probability mass of different OOD trajectories, changes throughout training. The bottom plot shows how the probability mass of the trajectories in our preference dataset (size 1) changes over training. The trajectories are listed in the legends for the plots, as a sequence of state, action pairs.

.

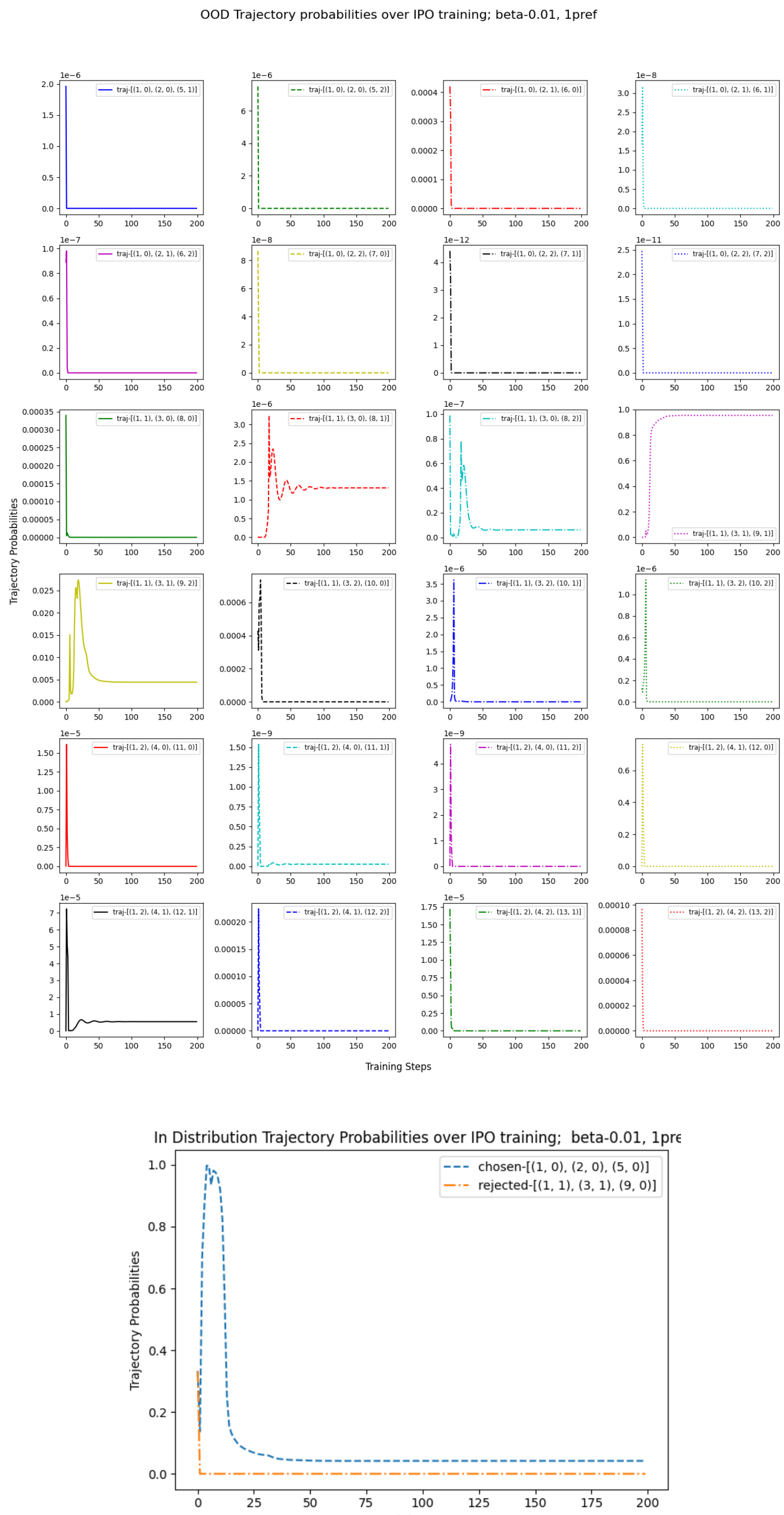

Figure 13: Trajectory probabilities throughout IPO training, $\beta = 0.01$. The top plot shows how the probability mass of different OOD trajectories, changes throughout training. The bottom plot shows how the probability mass of the trajectories in our preference dataset (size 1) changes over training. The trajectories are listed in the legends for the plots, as a sequence of state, action pairs.

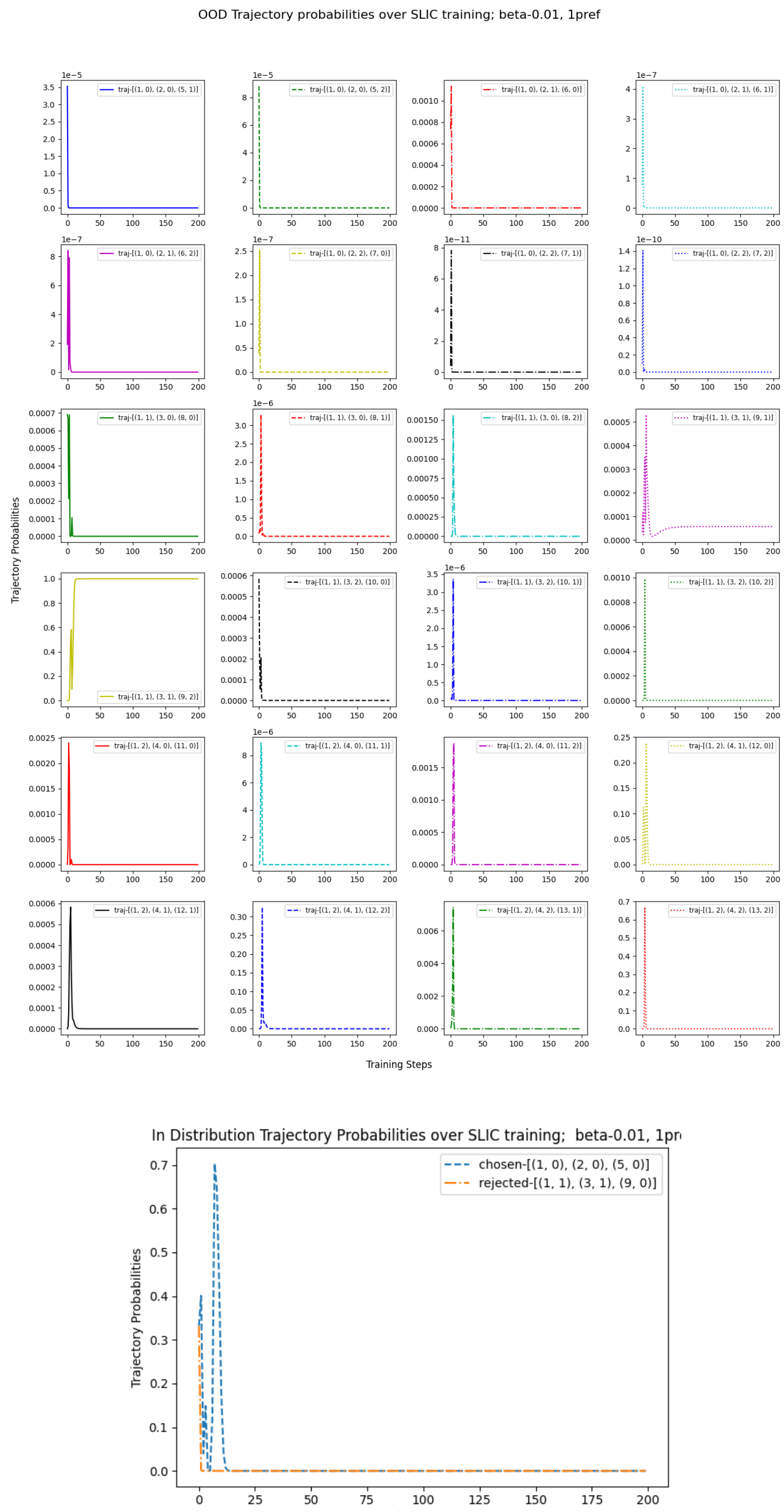

Figure 14: Trajectory probabilities throughout SLiC training, $\beta = 0.01$. The top plot shows how the probability mass of different OOD trajectories, changes throughout training. The bottom plot shows how the probability mass of the trajectories in our preference dataset (size 1) changes over training. The trajectories are listed in the legends for the plots, as a sequence of state, action pairs.

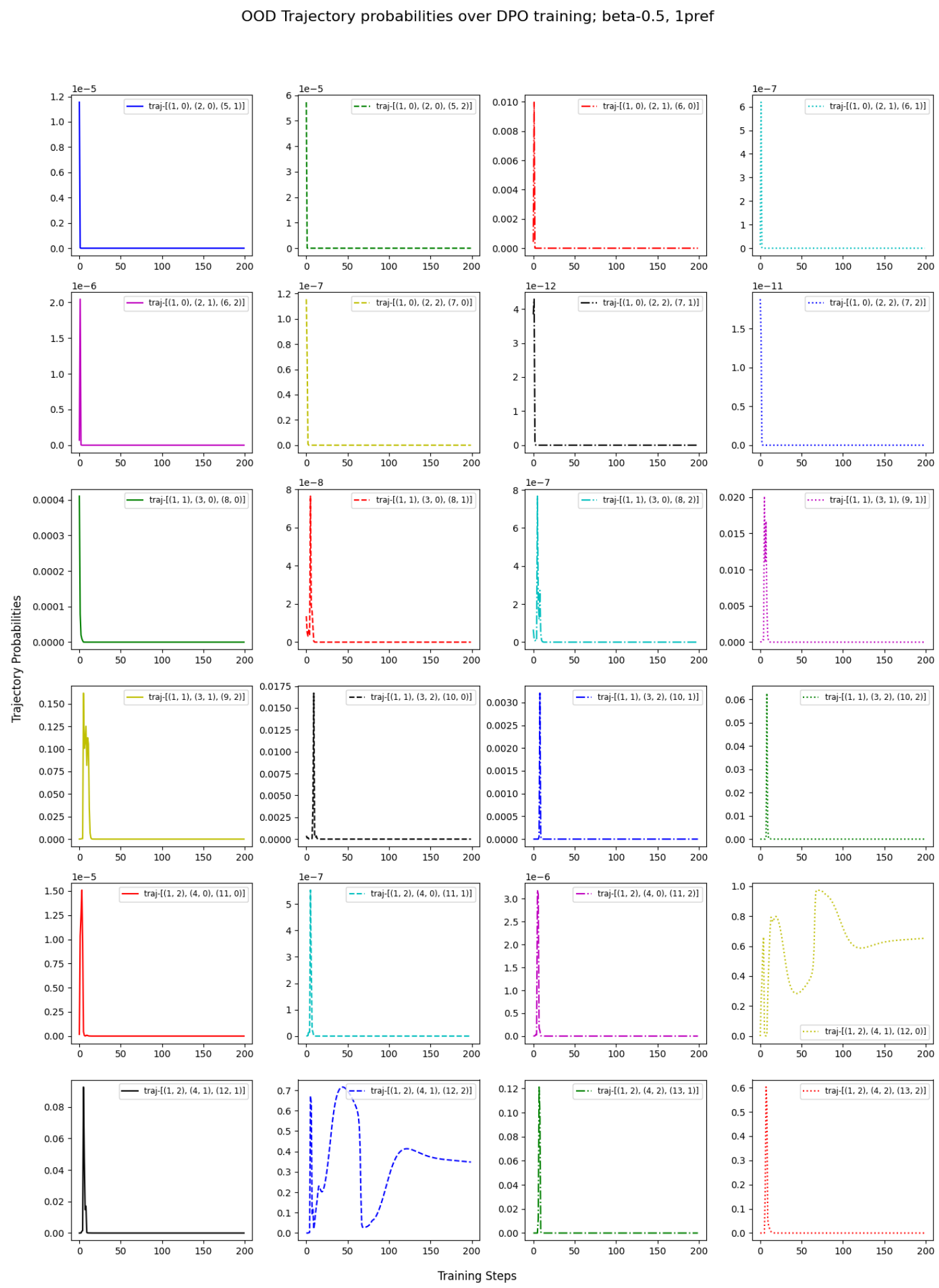

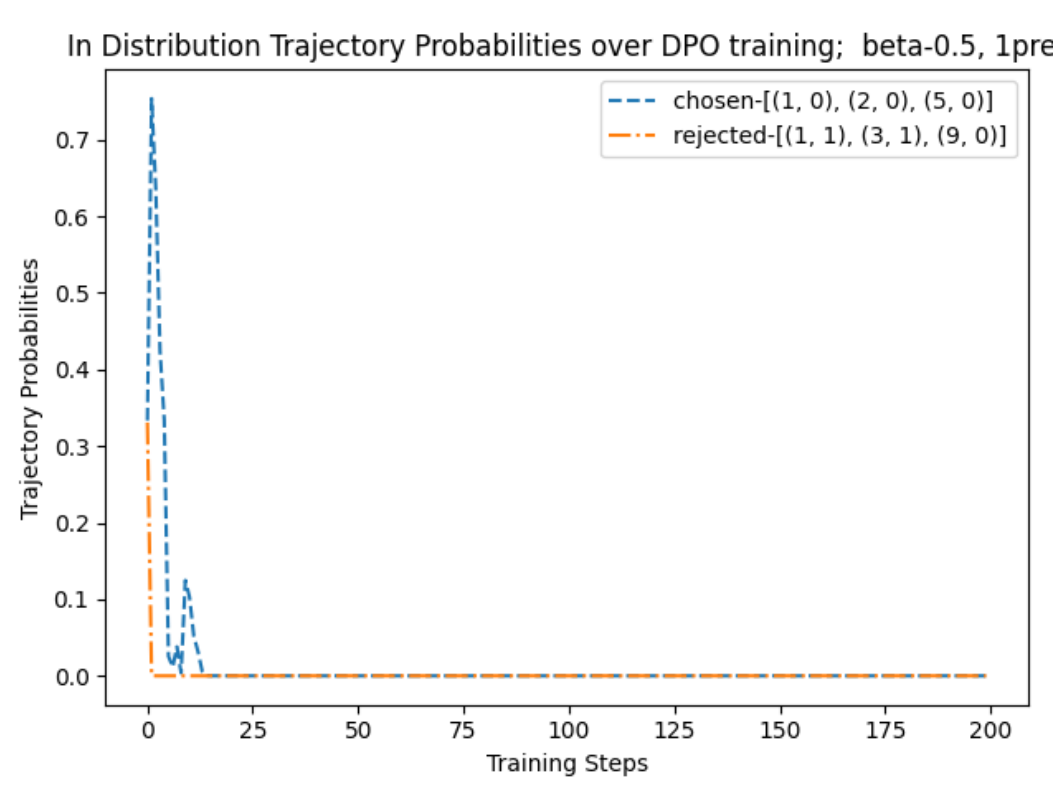

Figure 15: Trajectory probabilities throughout DPO training, $\beta = 0.5$. The top plot shows how the probability mass of different OOD trajectories, changes throughout training. The bottom plot shows how the probability mass of the trajectories in our preference dataset (size 1) changes over training. The trajectories are listed in the legends for the plots, as a sequence of state, action pairs.

.

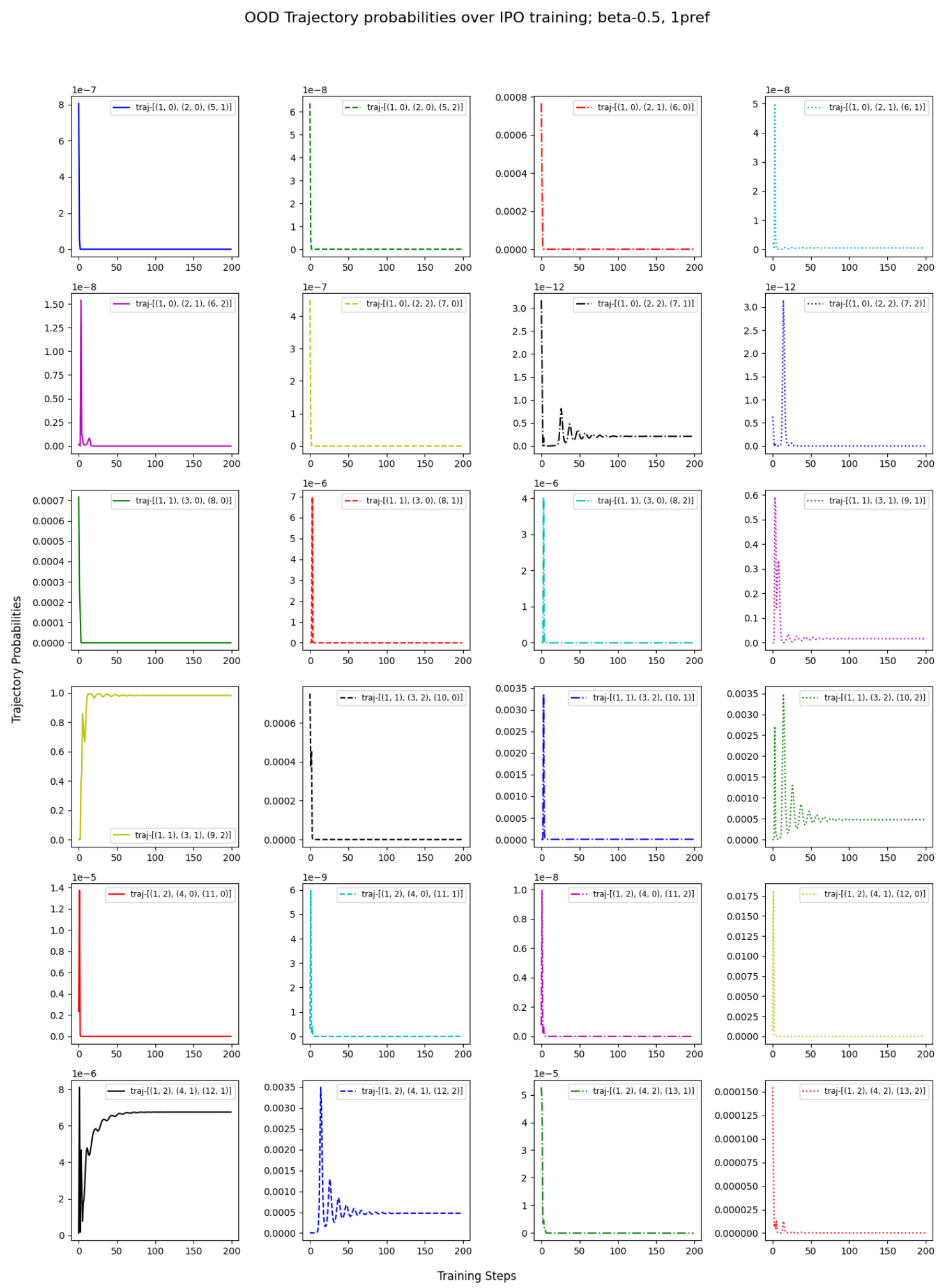

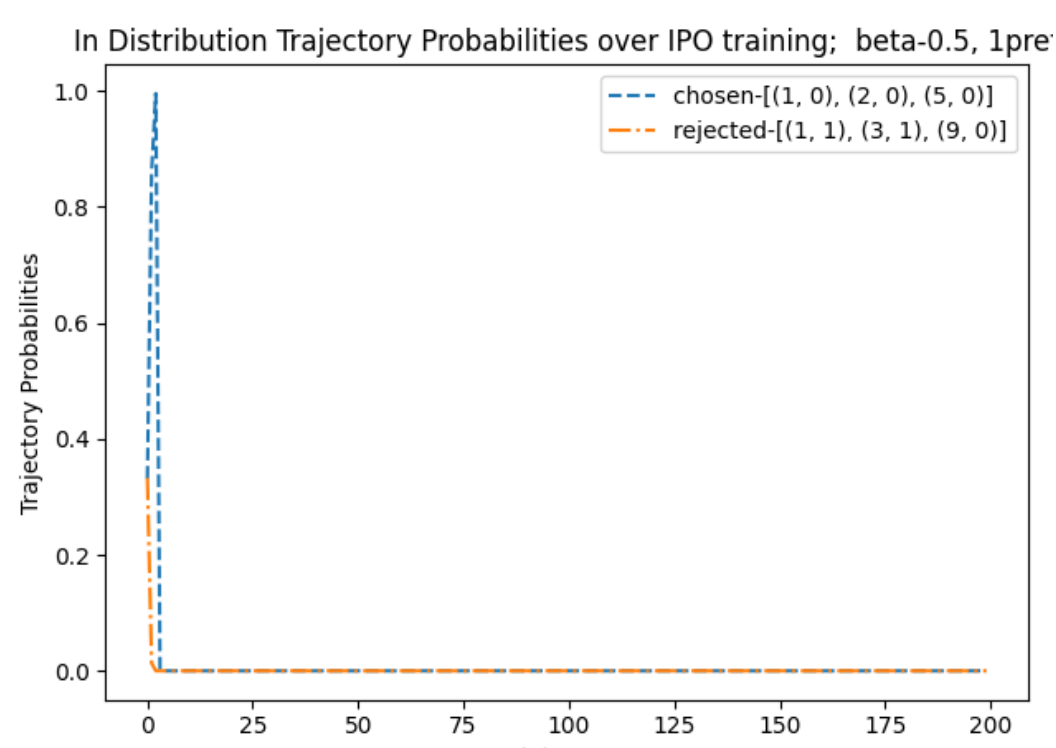

Figure 16: Trajectory probabilities throughout IPO training, $\beta = 0.5$. The top plot shows how the probability mass of different OOD trajectories, changes throughout training. The bottom plot shows how the probability mass of the trajectories in our preference dataset (size 1) changes over training. The trajectories are listed in the legends for the plots, as a sequence of state, action pairs.

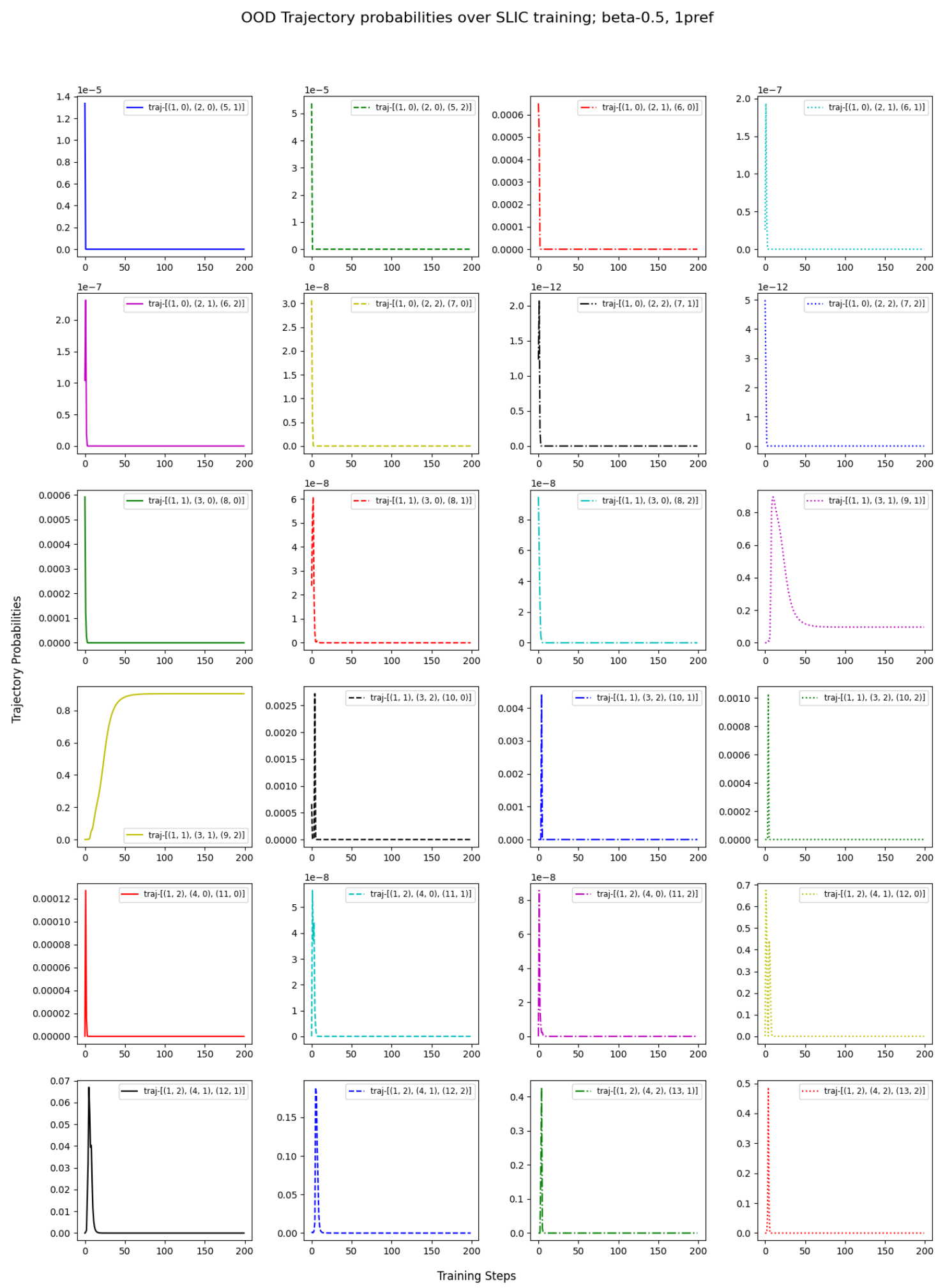

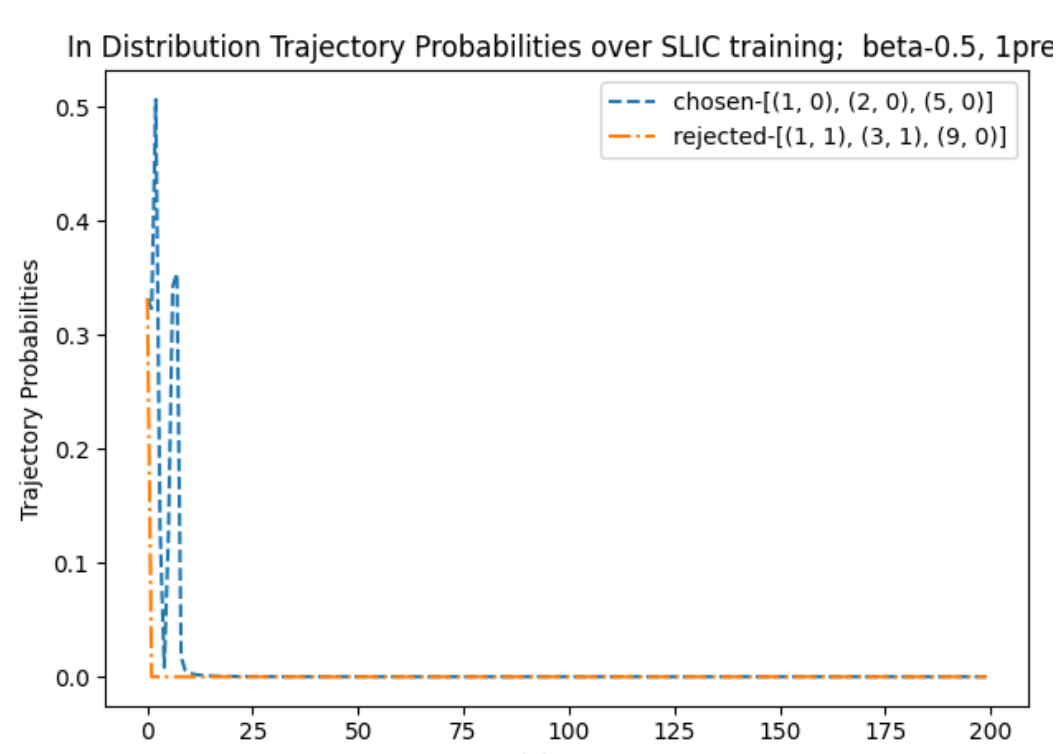

Figure 17: Trajectory probabilities throughout SLiC training, $\beta = 0.5$. The top plot shows how the probability mass of different OOD trajectories, changes throughout training. The bottom plot shows how the probability mass of the trajectories in our preference dataset (size 1) changes over training. The trajectories are listed in the legends for the plots, as a sequence of state, action pairs.

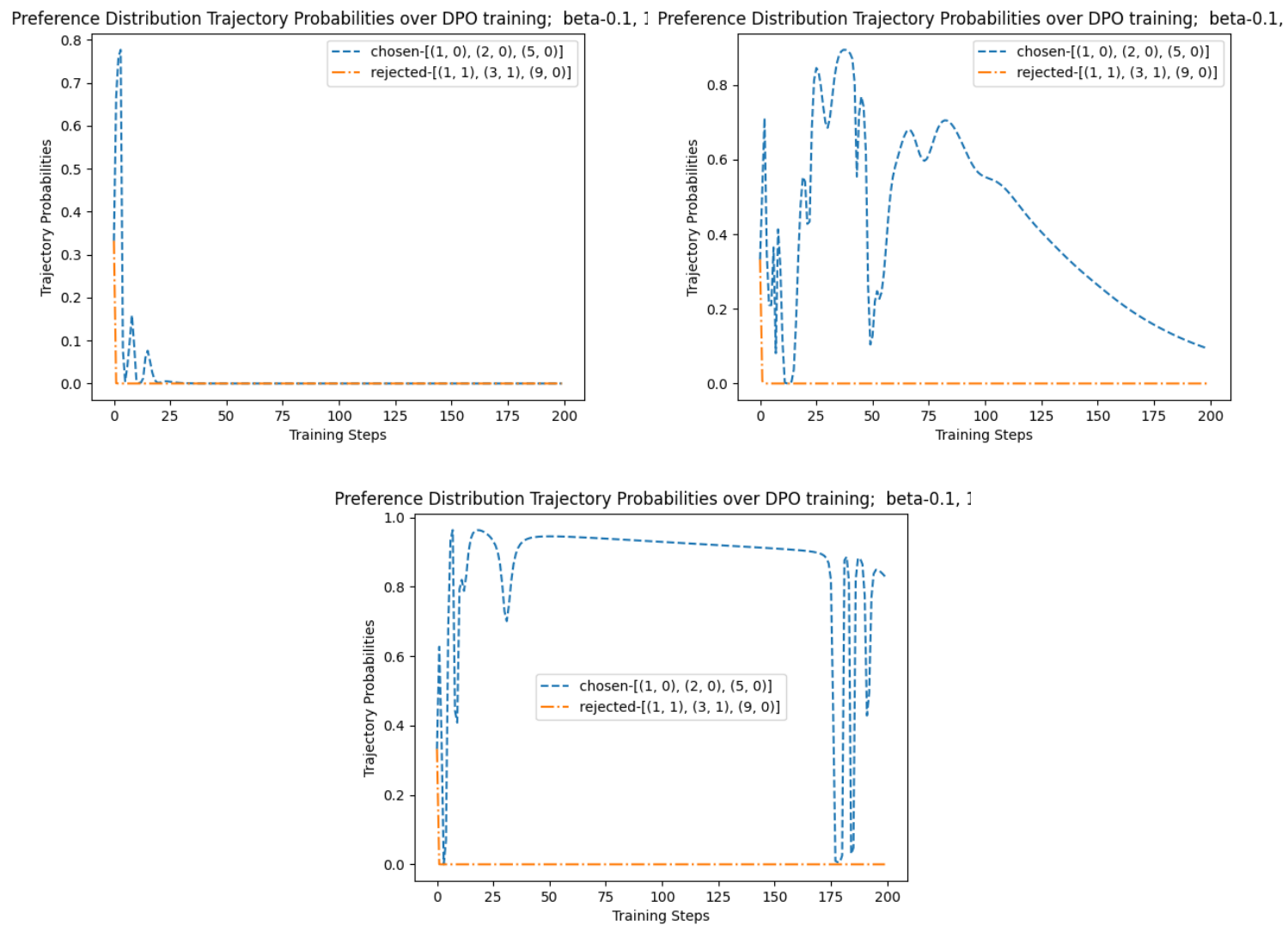

Figure 18: Trajectory probabilities throughout DPO training, over three different runs, with $\beta = 0.1$
.

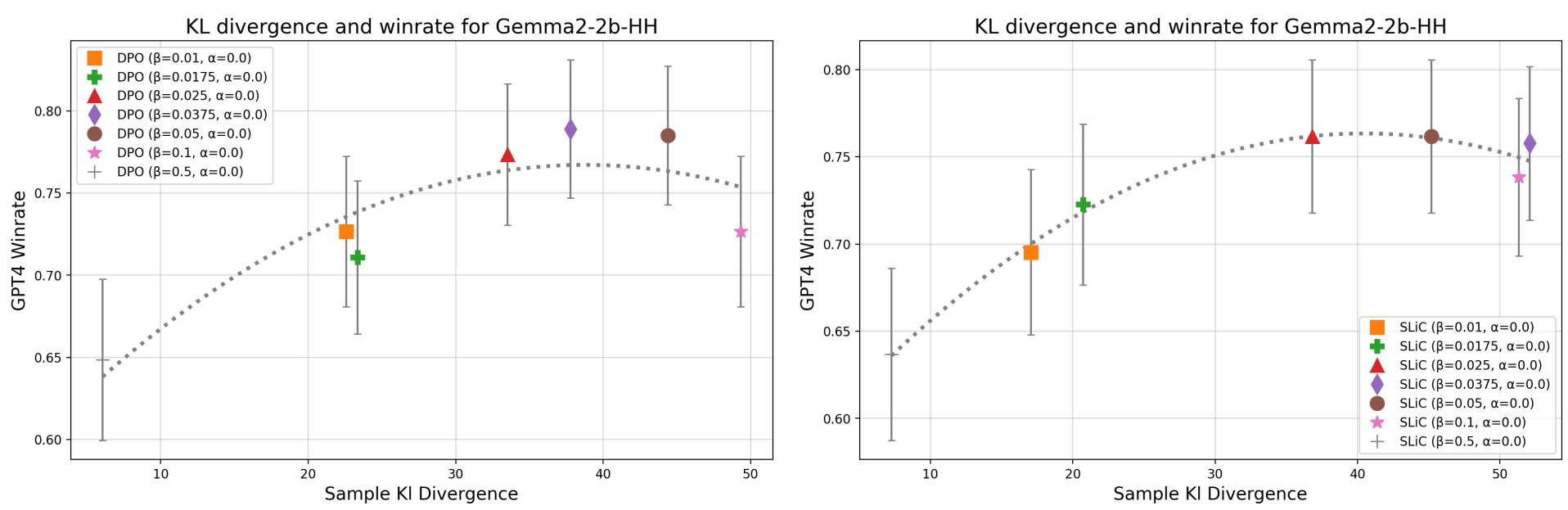

Figure 19: KL divergence versus GPT-4 win rate for the Gemma2-2b model on the Anthropic-HH dataset. The **left** plot shows DPO results, and the **right** plot shows SLiC results.

# F Overoptimization Trends in the Gemma2-2b Model and Anthropic-HH Dataset

We present KL divergence versus GPT-4 win rate plots in Figure 19 to illustrate overoptimization trends in Direct Alignment Algorithms for the Gemma2-2b model [61] and the Anthropic-HH dataset [5]. Results are shown for the DPO and SLiC variants, which sufficiently demonstrate that the overoptimization trends observed with the Pythia models are not specific to a single model or dataset. The figure illustrates the trade-off between KL divergence and GPT-4 win rate across different values of beta in the alignment objective.

