# OpenReview forum: "Scaling Laws for Reward Model Overoptimization in Direct Alignment Algorithms"
_NeurIPS.cc/2024/Conference — NeurIPS 2024 poster_

### Official Review · Reviewer_TztD · 2024-06-25

**Soundness:** 3
**Presentation:** 4
**Contribution:** 3
**Rating:** 7
**Confidence:** 4

**Summary:**

This paper analyse overoptimisation in the context of direct alignment algorithms (DAA). They show that even when no explicit reward model is being optimised against, a similar phenomena as Gao et al. is shown, where as KL budget increases "gold" reward increases and then decreases. This phenomena is shown across model sizes and different DAAs (DPO, IPO, SLiC). They fit scaling curves to this phenomena and find a similar style of fit to Gao et al.. This phenomena is then analysed from various angles, and the authors show that at low KL budgets best performance is reached early in training; length-correct adjusts the KL-winrate pareto frontier but doesn't mitigate overoptimisation; model training statistics aren't useful for predicting downstream performance. They also present some theoretical analysis and a toy MDP which demonstrates why reward exploitation can occur in DAAs, showing that lack of support in the sequence space combined with DAA's non-convex optimisation target can lead to DAAs placing high probability on OOD (and hence low-reward) sequences.

L. Gao, J. Schulman, and J. Hilton. Scaling laws for reward model overoptimization. Interna- tional Conference on machine Learning, 2023.

**Strengths:**

The paper's analysis is interesting and novel - while overoptimisation has been demonstrated in online RL preference learning algorithms, this demonstration of the effect in DAAs hasn't been presented before to my knowledge.

The quality of the analysis of the paper is high - they discuss their results neutrally and clearly, and the results are somewhat general. The additional analysis answers many obvious questions that come to mind, which is beneficial; the authors have clearly investigated this phenomena thoroughly.

The paper is well-presented and easy to understand.

The significance of the work is reasonably high - DAAs are a very popular class of preference learning algorithm, and this work has demonstrated a previously unknown limitation of these algorithms, and provided much additional analysis and understanding into how they work and where they can break, which will be useful for the community going forward in improving upon these algorithms and understanding their limitations.

**Weaknesses:**

## Large points

I think the main weaknesses with the paper are three-fold:
* All analysis is only performed on the TL;DR summarisation dataset, which is somewhat different from Gao et al. and the setting these preference learning algorithms are generaly used in (dialogue and instruction-following). While reproducing the whole analysis on another dataset is too much to ask, testing out some of the core hypotheses, or reproducing some of figure 1, on a different dataset (e.g. alpaca farm, or anthropic HH) would increase the robustness and generality of the results.
* Using GPT-4 winrate as the "gold" reward or reference output also makes the results less general. The preference distribution that produced the dataset the DAAs were trained on is not the same one being used to evaluate here. I would expect the results to hold up in the setting where these two distributions are the same, but it would be beneficial to have some empirical validation of that. This would likely mean mimicing the setting of Gao et al, and applying DPO on a preference distribution generated from an accessible gold reward function (for example, the data of Coste et al. is available and may be suitable, or alpaca farm gpt4 preference data could also work). The same reward function can then be used to produce the y-axis of the plots, which might even produce cleaner trends. This would more accurately mirror the real-world setting where human preferences would be used for both training and evaluation.
* While the dicussion in section 4 is useful, it would be beneficial to have a better explanation of why this phenomena happens in DAAs, as it's still not immediately intuitive for me.

## Small points

* it would be useful for the axes in figure 1 to be unified across all the subplots, so it is easier to compare between algorithms. Similarly for the y-axis in figure 2.
* some of the text in the figures (especially figure 7) is quite small.

## Summary

I think the paper is worthy of acceptance as is, and I'm reccomending accept. If one of the points mentioned in the Large Points section above was addressed thoroughly I would consider raising my score to a strong accept.

**Questions:**

My questions have been described in the weaknesses section.

**Limitations:**

The authors discuss limitations.

---

> ### Author Rebuttal · Authors · 2024-08-06
>
> We would like to thank the reviewer for the kind words!
>
> **All analysis is only performed on the TL;DR summarisation dataset, which is somewhat different from Gao et al. and the setting these preference learning algorithms are generaly used in (dialogue and instruction-following)...**
>
> We are working on replicating our main experiment with the Gemma-2 2B model on the Anthropic Helpful and Harmless dataset. Our initial results suggest similar over-optimization dynamics to the Pythia TL;DR experiments presented in the paper. We will include the additional experiment in our revised submission.
>
> **Using GPT-4 winrate as the "gold" reward or reference output also makes the results less general. The preference distribution that produced the dataset the DAAs were trained on is not the same one being used to evaluate here…**
>
> We do agree that the use of GPT-4 win rates as an evaluation metric makes the results more noisy, but we believe this is a more realistic scenario.
> The TL;DR dataset is based on human rater feedback. In original DPO work [3] the authors carried out a human evaluation study on the TL;DR summarization task and found GPT-4 - human agreement to be 67-70%, which was higher than intra-human annotator agreement of 65%.
>
> **While the dicussion in section 4 is useful, it would be beneficial to have a better explanation of why this phenomena happens in DAAs, as it's still not immediately intuitive for me.**
>
> We will expand section 4 to try and make it more clear given additional space for the final paper.
> Fundamentally, DAAs can be viewed as performing a type of generalized linear model, modulated by a convex regularization function $g$, where the number of datapoints $N$ is much much smaller than the number of features (prompt-response space). Normally, one would apply some type of regularization (like L2 in ridge regression) to make the problem strictly convex, but DAAs don’t do that. Instead, we are left with a vastly under-constrained problem. Because of this “rank-deficiency”, there are a large number of solutions that place probability on out-of-distribution sequences. Thus, DAA methods can easily start to converge to one of these solutions during training.
> A simple construction is as follows: consider a setting where the only prompt space is empty (ie no prompt) and the response is one of three tokens: (a, b, c). If my preference dataset does not include c and contains at least one conflicting preference (a > b and b < a), then the minima of a DAA will just ensure that the log-ratio of a and b equal some finite value. Since only the ratio of a and b is enforced at the optima, we can place any amount of probability mass on response c.
> We have also detailed sufficient conditions for it to occur in response to Reviewer bFR8.
> We would appreciate it if the reviewer could let us know what we could do to help make this section more understandable.
>
>
>
> [1] Direct Preference Optimization: Your Language Model is Secretly a Reward Model, Rafael Rafailov, Archit Sharma, Eric Mitchell, Stefano Ermon, Christopher D. Manning, Chelsea Finn

---

> ### Comment · Reviewer_TztD · 2024-08-10
>
> Thank you for your response. I'm glad to here you're working on replicating the results in a different setting, and the preliminary results are very promising in this regard.
>
> ### on gpt-4 winrate
>
> The scenario we care about is when we use humans to produce preferences, which we then optimise against using some DAA algorithm, and then we see that we have overoptimised the human preference function. Given it's expensive to do this analysis with real humans, we can produce an analogous setting where we replace humans with GPT4 as the rater. However, I think to make this setting more analagous we would like to have GPT4 both as the provider of preferences and as the as the judge used during evaluation, whereas in your experiments humans provide the preferences but GPT4 is the judge, resulting in a mismatch and a potentially less analogous setting. This is the issue with the experimental settings I was pointing to in that comment.
>
> I acknowledge reproducing the experimental results with GPT-4 as the provider of preferences as well as the rater would be extremely expensive and infeasible in the remaining time, but I think it would be worth discussing this disanalogy between your setting and the real-world scenario that motivates this work in the paper.
>
> ### on better intuition
>
> Thanks for the providing that explanation. I understand that intuition, but to me it doesn't seem that the explanation provided predicts the shape of the relationship between KL and gold reward. As described, that intuition would imply (to me) that as soon as you start optimising with the DAA method, KL would increase and gold reward would go down. Why do you think gold reward goes up and then down as you optimise (or as you choose different KL penalty coefficients)?
>
> ### summary
>
> Overall, I am mainting my score of a 7, but I am keen to see the paper accepted, especially given the preliminary results reproduce the effect in a different setting.

---

> > ### Author Response · Authors · 2024-08-11
> >
> > Regarding the evaluation using GPT4, we do agree that in theory this represents a distribution shift in terms of preferences. However, we would like to highlight the below table from the original DPO work:
> >
> > |                           | DPO | SFT | PPO-1 |
> > |---------------------------|-----|-----|-------|
> > | **N respondents**          | 272 | 122 | 199   |
> > | **GPT-4 (S) win %**        | 47  | 27  | 13    |
> > | **GPT-4 (C) win %**        | 54  | 32  | 12    |
> > | **Human win %**            | 58  | 43  | 17    |
> > |---------------------------|-----|-----|-------|
> > | **GPT-4 (S)-H agree**      | 70  | 77  | 86    |
> > | **GPT-4 (C)-H agree**      | 67  | 79  | 85    |
> > | **H-H agree**              | 65  | -   | 87    |
> >
> > It shows that specifically on the TL;DR summarization task, GPT4 agrees with the majority opinion at the same degree (and higher for DPO models) as humans. We would argue that GPT4 judgments are as good of a proxy for the majority human opinion as individual human raters (which is the way the training data was generated).
> >
> > We do agree that the drivers behind the exact KL-quality dynamics are still somewhat unclear. However, we believe this is a challenging theoretical and empirical problem, which may warrant it's own research as some recent works have done [1].
> >
> > [1] Understanding the Learning Dynamics of Alignment with Human Feedback
> > Shawn Im, Yixuan Li

---

### Official Review · Reviewer_bFR8 · 2024-07-04

**Soundness:** 3
**Presentation:** 3
**Contribution:** 3
**Rating:** 6
**Confidence:** 4

**Summary:**

This paper studies the reward over-optimization issue for offline alignment algorithms (e.g., DPO series) with massive experiment trials and discussions on why this phenomenon happens.

**Strengths:**

1. This is the first paper that studies the over-optimization issue for DPO-like algorithms systematically. The observations and findings can be beneficial to the community. The proposed experiments are organized rigorously with sufficient quantitative results.

2. The paper presents concrete examples to reveal why over-optimizations could happen for DPO-like algorithms, which brings insights to the community.

**Weaknesses:**

There are no major weaknesses. Here are some of the minor comments for further improvement.

1. Why do you choose DPO, IPO, and SLiC-HF? Why not some other variants, say ORPO? Although I do agree that the current selections can be sufficiently representative, some discussions on why they are prioritized can be appreciated.

2. The presentation of Section 4 can be possibly improved. With its current presentation, it is difficult for a reader to understand the major argument of this section. There seems to be a collection of diverse arguments with different experiments. It may be better to highlight the overall goal of this section and the reason why these experiments are conducted in the very beginning before moving into more detailed experiments.

3. Regarding the "Rank Efficiency" issue in Section 4, is it possible to have some more rigorous formulations in addition to statements only, say a theorem?

4. Regarding citations and references, the OOD issue of DPO is also discussed in [1], which may be a good one to refer to. Also, the citation format is very strange. For example, many cited papers in the reference do not have any publication venues.

[1] Is DPO Superior to PPO for LLM Alignment? A Comprehensive Study, S. Xu et al., ICML 2024. https://arxiv.org/abs/2404.10719

**Questions:**

See the comments in the weakness section.

**Limitations:**

The limitations have been discussed in the paper.

---

> ### Author Rebuttal · Authors · 2024-08-06
>
> **Why do you choose DPO, IPO, and SLiC-HF? Why not some other variants, say ORPO? Although I do agree that the current selections can be sufficiently representative, some discussions on why they are prioritized can be appreciated.**
>
> We chose these versions of the DAAs as they were the most extensively studied in prior literature and used in practice at the time of our writing. We agree with the reviewer that it would be useful to expand the algorithm selection, which we could not do due to limited resources.
> The ORPO algorithm specifically does not use a reference SFT model, but directly aligns the base model with the feedback data. This makes it harder to fit in the same performance-divergence framework.
>
> **The presentation of Section 4 can be possibly improved. With its current presentation, it is difficult for a reader to understand the major argument of this section. There seems to be a collection of diverse arguments with different experiments. It may be better to highlight the overall goal of this section and the reason why these experiments are conducted in the very beginning before moving into more detailed experiments.**
>
> We believe the reviewer might be referring to Section 3. The goal of that section is to study the potential causes and dynamics of over-optimization, i.e. training objective (type of DAAA), capacity (model size), spurious correlations (length exploitation experiments) etc.. We will add language explaining this at the beginning and how each experiment fits within that framework.
>
> **Regarding the "Rank Efficiency" issue in Section 4, is it possible to have some more rigorous formulations in addition to statements only, say a theorem?**
>
>
> Yes, we will present a more rigorous construction in final draft, which we will outline here. Consider the “query” vectors from section 4, which select the “win” and “loss” prompt-response pairs with a value of +1 or -1 from the prompt-response space, $X \times Y$. As shown in the Appendix of Hejna et al. [1]  there are two sufficient conditions for the rank deficiency issue:
>
> 1.  if the null space of the matrix formed by these query vectors $Q \in \mathbb{R}^{N \times |X \times Y|}$ is non-trivial on the support of the preference dataset (i.e. there exists a vector $v \in N(Q)$ such that $v(x,y) \ne 0$
>
> 2. If for every prompt $x$, there is a response $y$ not included in the preference dataset.
>
> The first condition is easily satisfied when the preference dataset contains any disagreeing preferences. The second is easily satisfied when the size of the preference dataset is smaller than the prompt response space, which is almost always the in practice as the prompt-response space is exponential in length. A trivial example of this is if the prompt space is empty and the response space has three tokens (a, b, c). If the preference dataset is (a < b, b > a) then the DAA loss is minimized by any policy that places equal probability on a and b, even if the probability of c is highest.
>
>
> **Regarding citations and references, the OOD issue of DPO is also discussed in [1], which may be a good one to refer to. Also, the citation format is very strange. For example, many cited papers in the reference do not have any publication venues.**
>
> Thank you for bringing this to our attention, we will add the citation and rectify the formatting in our updated version!
>
> [1] Contrastive Preference Learning: Learning from Human Feedback without RL, Joey Hejna, Rafael Rafailov, Harshit Sikchi, Chelsea Finn, Scott Niekum, W. Bradley Knox, Dorsa Sadigh

---

> > ### Author Response · Authors · 2024-08-12
> >
> > Dear reviewer,
> >
> > We would like to follow up to see if the response addresses your concerns or if you have any further questions. We would really appreciate the opportunity to discuss this further if our response has not already addressed your concerns. Thank you again!

---

> > > ### Comment · Reviewer_bFR8 · 2024-08-12
> > > **Thanks**
> > >
> > > The responses all look good to me.

---

### Official Review · Reviewer_r4HU · 2024-07-13

**Soundness:** 2
**Presentation:** 3
**Contribution:** 2
**Rating:** 5
**Confidence:** 4

**Summary:**

This paper studies the scaling laws of DAAs for RLHF. The authors conducted extensive empirical studies and the discoveries are reported and discussed.

**Strengths:**

The paper is well-written and easy to follow.
The authors conducted extensive empirical studies.
The current results are useful to the community to understand different algorithms.
The design of the diagnostic toy MDP is interesting and smart.

**Weaknesses:**

I like large-scale empirical study yet I disagree with calling those discoveries "laws". To draw a clear scaling **law** (from a scientific perspective), the current results are not supportive enough. Specifically, in Figure 1, the results do not seem to be a good fit, and more experimental results might be helpful to draw the conclusion. Also, why is the form of "scaling law" chosen as Eqn(5)? In Gal et al, BoN and PPO have different forms --- it would be highly possible that different DAAs also have different equation forms.


The authors should further highlight the takeaway messages of their empirical discoveries. For instance, with the discovered scaling law, is there a way to perform early stopping in training to achieve better performance? --- This could be a great contribution to the community as otherwise researchers may be prone to make unfair comparisons in evaluating algorithms.

**Questions:**

Could the authors also provide error bars in Figure 2, second row? The results look very unstable.

For the scaling law fit, the author mentioned using win-rate to be the proxy of the golden reward, could the author provide detailed statistics of the results? I'm curious --- to what extent could different judges agree? If the win rate is only accurate in (for example,) 80% of the settings, does not that mean we are only able to draw a very rough conclusion using another proxy of this objective?

On the Tree-MDP, why should all states go to the same absorbing state? What is the consideration of using a single absorbing state rather than many (i.e., = the number of leaves).

**Limitations:**

please see weakness / question.

---

> ### Author Rebuttal · Authors · 2024-08-06
>
> We would like to thank the reviewer for the useful feedback!
>
> **Specifically, in Figure 1, the results do not seem to be a good fit, and more experimental results might be helpful to draw the conclusion.**
>
> We used win rates as computed by GPT-4 for evaluation, which is now well-established in the “LLM-as-a-judge” framework to be a decent proxy for model quality as evaluated by humans [1]. The reviewer is indeed right that this approach could inject some noise in the evaluations.
> We do agree that further evaluations, such as higher number of evaluation samples (we used 256 held-out prompts), more training runs with different KL-parameter exploration and more intermediate checkpoints would reduce noise and draw a stronger statistical dependency. However, these require significant additional resources, both computational for model training as well as credits for LLM evaluations.
>
> **Also, why is the form of "scaling law" chosen as Eqn(5)?**
>
> We evaluated a number of statistical formulations of the scaling law and found the ones presented in Gao et. al. [5] to provide as good statistical fit as any other formulation across the board. It is however possible that our search was not exhaustive and a function of similar complexity can better fit the data.
>
> **The authors should further highlight the takeaway messages of their empirical discoveries. For instance, with the discovered scaling law, is there a way to perform early stopping in training to achieve better performance? --- This could be a great contribution to the community as otherwise researchers may be prone to make unfair comparisons in evaluating algorithms.**
>
> The goal of our work is to highlight the empirical phenomenon of reward over-optimization in DAAs, which had not been studied before as far as we are aware. We hope our work can incentivize a broader research direction of robustness for DAAs, similar to the line of research that the  Gao et. al. [5] publication spurred.
> There are a number of promising directions to pursue on that front, such as model merging for example [2] or reward smoothing for DAAs [6]. We hope our work will incentivize researchers to pursue such ideas in follow-up publications. We ourselves are investigating mitigating issues, which we believe warrant independent investigations.
>
> **Could the authors also provide error bars in Figure 2, second row? The results look very unstable.**
>
> We will include the modified graph in our updated camera-ready submission.
>
> **For the scaling law fit, the author mentioned using win-rate to be the proxy of the golden reward, could the author provide detailed statistics of the results? I'm curious --- to what extent could different judges agree? If the win rate is only accurate in (for example,) 80% of the settings, does not that mean we are only able to draw a very rough conclusion using another proxy of this objective?**
>
> In the original DPO work [3], the authors carried out a human evaluation study on the TL;DR summarization task and found GPT-4 - human agreement to be 67-70%, which was higher than intra-human annotator agreement of 65%. A number of prior works have also studied this setting extensively [1], [4] and have established the use of “LLM-as-a-judge”.
>
> **On the Tree-MDP, why should all states go to the same absorbing state? What is the consideration of using a single absorbing state rather than many (i.e., = the number of leaves).**
>
> The absorbing state in the toy-MDP is used as an analogue to the end-of-sequence token in order to faithfully reflect the true LLM finetuning setting. Theoretically, the absorbing state does not affect optimization as all actions from the pre-absorbing state leads to the absorbing state with 1 probability and its effect is canceled across the preference pairs and reference policy.
>
> [1] AlpacaFarm: A Simulation Framework for Methods that Learn from Human Feedback, Yann Dubois, Xuechen Li, Rohan Taori, Tianyi Zhang, Ishaan Gulrajani, Jimmy Ba, Carlos Guestrin, Percy Liang, Tatsunori B. Hashimoto
>
> [2] WARP: On the Benefits of Weight Averaged Rewarded Policies, Alexandre Ramé, Johan Ferret, Nino Vieillard, Robert Dadashi, Léonard Hussenot, Pierre-Louis Cedoz, Pier Giuseppe Sessa, Sertan Girgin, Arthur Douillard, Olivier Bachem
>
> [3] Direct Preference Optimization: Your Language Model is Secretly a Reward Model, Rafael Rafailov, Archit Sharma, Eric Mitchell, Stefano Ermon, Christopher D. Manning, Chelsea Finn
>
> [4] Judging LLM-as-a-Judge with MT-Bench and Chatbot Arena, Lianmin Zheng, Wei-Lin Chiang, Ying Sheng, Siyuan Zhuang, Zhanghao Wu, Yonghao Zhuang, Zi Lin, Zhuohan Li, Dacheng Li, Eric P. Xing, Hao Zhang, Joseph E. Gonzalez, Ion Stoica
>
> [5] Scaling Laws for Reward Model Overoptimization, Leo Gao, John Schulman, Jacob Hilton
>
> [6] Iterative Data Smoothing: Mitigating Reward Overfitting and Overoptimization in RLHF, Banghua Zhu, Michael I. Jordan, Jiantao Jiao

---

> > ### Author Response · Authors · 2024-08-12
> >
> > Dear reviewer,
> >
> > We would like to follow up to see if the response addresses your concerns or if you have any further questions. We would really appreciate the opportunity to discuss this further if our response has not already addressed your concerns. Thank you again!

---

### Official Review · Reviewer_Q9V7 · 2024-07-27

**Soundness:** 3
**Presentation:** 3
**Contribution:** 3
**Rating:** 6
**Confidence:** 4

**Summary:**

Reinforcement Learning from Human Feedback (RLHF) is a popular paradigm for aligning Large Language Models (LLMs) to human preferences. Direct Alignment Algorithms (DAA) are an alternative to traditional RLHF methods, which reduce the need to learn a reward and policy model separately. It has been shown that traditional RLHF methods suffer from reward over-optimization or reward hacking. However, because DAA do not learn a separate reward model, it is important to study whether these algorithms suffer from the same issues. This paper studies three popular DAA algorithms and how their performance deteriorates from over-optimization at scale. The paper has several compelling results that show the trade-off between different DAA objectives, KL divergence, and win rate when optimizing across different scales of models.

**Strengths:**

- Empirical experiments provide insight into DAA algorithm's behavior at scale.
- The empirical verification of the decreasing likelihood and model performance provides insight into an important issue in the literature: the decrease in the responses of both chosen and rejected samples.
- The paper is generally well-written, and I was able to follow most empirical conclusions.

**Weaknesses:**

- The paper only performs experiments on a single model class and task, so it is not clear if these results generalize. For example, the paper observes that most models perform best after training on only 25% of the data, but it is not obvious if this is an artifact of the specific models used or the data. Does the same observation hold true on a different model or a different task?
- The performance of DPO seems under-tuned compared to the results of DPO in the literature. (1, 2).
- The paper does not provide any details regarding how model selection was performed and additional design choices.

(1) REBEL: Reinforcement Learning via Regressing Relative Rewards by Zhaolin Gao

(2) BPO: Supercharging Online Preference Learning by Adhering to the Proximity of Behavior LLM

**Questions:**

- Could you provide best-fit lines for each model size in Figure 4 so that I can see trends? (This is the point raised on lines 196-200.)
- Is it a fair conclusion that IPO performs the best among the DAA studied in this paper?
- Are IPO and SCLI also prone to length exploitation? If so, could you provide results similar to those for DPO?
- How did you perform model selection?

**Limitations:**

Yes

---

> ### Author Rebuttal · Authors · 2024-08-06
>
> We would like to thank the reviewer for the informative review!
>
> **The paper only performs experiments on a single model class and task, so it is not clear if these results generalize…**
>
> We are working on replicating our main experiment with the Gemma-2 2B model on the Anthropic Helpful and Harmless dataset. Our initial results suggest similar over-optimization dynamics to the Pythia TL;DR experiments presented in the paper. We will include the additional experiment in our revised submission.
>
>
>
> **The performance of DPO seems under-tuned compared to the results of DPO in the literature. (1, 2).**
>
> These discrepancies are likely due to some parameter choices. For example REBEL (1) uses generation temperature of 0.1 and max-token length of 53, while we use sampling temperature 1.0 and max-token length of 512 for all of our experiments.  This can affect performance on TL;DR as shown in the original DPO work [1]. There is also a slight difference in the wording of the evaluation prompt as we also use “concise” in our evaluation prompt, which may lower win-rates for longer summaries. We will include all of these details in an updated appendix with our camera-ready version.
> We could not find enough details on these parameters in the second reference.
>
>
>
> **The paper does not provide any details regarding how model selection was performed and additional design choices.**
>
> Could the reviewer expand on this point? We report performance of the final trained checkpoint after 1 epoch of DAA training, as well as intermediate checkpoints at regular intervals. We do not carry out any more involved checkpoint selection.
>
>
>
> **Could you provide best-fit lines for each model size in Figure 4 so that I can see trends? (This is the point raised on lines 196-200.)**
>
> We will include those modifications in our updated camera-ready submission.
>
>
>
> **Is it a fair conclusion that IPO performs the best among the DAA studied in this paper?**
>
> For all the algorithms we studied (DPO, IPO, SLiC) the best checkpoints reach similar performance. However, IPO indeed seems to be more robust to the over-optimization phenomenon.
>
>
>
> **Are IPO and SCLI also prone to length exploitation? If so, could you provide results similar to those for DPO?**
>
> We have carried out additional evaluations on this issue. All the studied algorithms show significant length increases in the response, beyond the dataset coverage. However the relationship between length and implicit rewards seems to be more nuanced than the results shown in Figure 3 for DPO. We will include these additional experiments in our updated camera-ready submission.
>
> [1] Direct Preference Optimization: Your Language Model is Secretly a Reward Model
> Rafael Rafailov, Archit Sharma, Eric Mitchell, Stefano Ermon, Christopher D. Manning, Chelsea Finn

---

> > ### Author Response · Authors · 2024-08-12
> >
> > Dear reviewer,
> >
> > We would like to follow up to see if the response addresses your concerns or if you have any further questions. We would really appreciate the opportunity to discuss this further if our response has not already addressed your concerns. Thank you again!

---

> > > ### Comment · Reviewer_Q9V7 · 2024-08-12
> > >
> > > Thank you for the clarification, I was not sure if you did any involved model checkpoint selection. All of my concerns have been addressed.

---

### Author Rebuttal · Authors · 2024-08-07

We would like to thank all the reviewers for the useful comments!

1. We are working to expand the experiments in our work using the Gemma 2 2B model on the Anthropic Helpful and Homeless Dataset. We have attached our preliminary results here, which show the same general effects as our Pythia TL;DR experiments. We will include the final set of experiments in our camera-ready submissions.

2. While we do not use the formulation of gold reward models, we use win rates as evaluated by GPT-4, which we believe is a more realistic evaluation. Prior work [1] has shown that GPT-4 achieves higher average agreement with human annotators than they do between themselves on the TL;DR summarization task.
We believe this makes our setting and results a good proxy evaluation for real human preference, which a gold reward model was designed to approximate in the original Gao et. al. work [2].

3. We have attached the additionally requested analysis and figures on individual model fits and length correlations, which will also be included in our final camera-ready submission.

4. We will expand on our theoretical formulation of Section 4  as outlined in individual reviewer responses.

[1] Direct Preference Optimization: Your Language Model is Secretly a Reward Model, Rafael Rafailov, Archit Sharma, Eric Mitchell, Stefano Ermon, Christopher D. Manning, Chelsea Finn

[2] Scaling Laws for Reward Model Overoptimization, Leo Gao, John Schulman, Jacob Hilton

---

### Decision · Program_Chairs · 2024-09-25

**Decision:**

Accept (poster)

**Comment:**

This paper studies Direct Alignment Algorithms (DAAs), revealing that similar to traditional RLHF methods, DAAs suffer from degradation due to over-optimization, especially at higher KL-budgets. The authors also examine the consequences of reward over-optimization in DAAs across various objectives, training regimes, and model scales. The results seem to be interesting for a broad audience of NeurIPS. One issue to be careful about, as pointed out by "Reviewer TztD": there is a potential distribution shift when evaluation using GPT-4. I strongly recommend that the authors add a discussion on this point in the paper.